# Electric control of a canted-antiferromagnetic Chern insulator

Jiaqi Cai[1,6], Dmitry Ovchinnikov[1,6], Zaiyao Fei[1], Minhao He[1], Tiancheng Song[1], Zhong Lin[1], Chong Wang[1,2], David Cobden[1], Jiun-Haw Chu[1], Yong-Tao Cui[3], Cui-Zu Chang[4], Di Xiao[1,2], Jiaqiang Yan[5] & Xiaodong Xu[1,2✉]

The interplay between band topology and magnetism can give rise to exotic states of matter. For example, magnetically doped topological insulators can realize a Chern insulator that exhibits quantized Hall resistance at zero magnetic field. While prior works have focused on ferromagnetic systems, little is known about band topology and its manipulation in antiferromagnets. Here, we report that $MnBi_2Te_4$ is a rare platform for realizing a canted-antiferromagnetic (cAFM) Chern insulator with electrical control. We show that the Chern insulator state with Chern number $C = 1$ appears as the AFM to canted-AFM phase transition happens. The Chern insulator state is further confirmed by observing the unusual transition of the $C = 1$ state in the cAFM phase to the $C = 2$ orbital quantum Hall states in the magnetic field induced ferromagnetic phase. Near the cAFM-AFM phase boundary, we show that the dissipationless chiral edge transport can be toggled on and off by applying an electric field alone. We attribute this switching effect to the electrical field tuning of the exchange gap alignment between the top and bottom surfaces. Our work paves the way for future studies on topological cAFM spintronics and facilitates the development of proof-of-concept Chern insulator devices.

---

[1] Department of Physics, University of Washington, Seattle, WA 98195, USA. [2] Department of Materials Science and Engineering, University of Washington, Seattle, WA 98195, USA. [3] Department of Physics and Astronomy, University of California, Riverside, CA 92521, USA. [4] Department of Physics, The Pennsylvania State University, University Park, PA 16802, USA. [5] Materials Science and Technology Division, Oak Ridge National Laboratory, Oak Ridge, TN 37831, USA. [6] These authors contributed equally: Jiaqi Cai, Dmitry Ovchinnikov. ✉email: xuxd@uw.edu

A Chern insulator is a two-dimensional topological state of matter with quantized Hall resistance of $h/Ce^2$ and vanishing longitudinal resistance[1,2], where the Chern number $C$ is an integer that determines the number of topologically protected chiral edge channels[1,3–5]. The formation of the Chern insulator requires time-reversal symmetry breaking, which is usually achieved by magnetic doping or magnetic proximity effect[3,6,7]. A seminal example is the magnetically-doped topological insulators that realize a ferromagnetic (FM) Chern insulator with the quantum anomalous Hall effect[2]. Although Chern insulators have now been realized in several systems, little is known about this topological phase in antiferromagnets, which may offer a platform for exploring new physics and control of band topology[8,9].

MnBi$_2$Te$_4$, an intrinsic topological magnet, provides a new platform for incorporating band topology with different magnetic states[10–14]. MnBi$_2$Te$_4$ is a layered van der Waals compound that consists of Te–Bi–Te–Mn–Te–Bi–Te septuple layers (SL) stacked along the crystallographic $c$-axis. At zero magnetic field, it hosts an A-type antiferromagnetic (AFM) ground state: each SL of MnBi$_2$Te$_4$ individually exhibits ferromagnetism with out-of-plane magnetization, while the adjacent SLs couple antiferromagnetically[13,14]. By applying an external magnetic field perpendicular to the SLs, the magnetic state evolves from AFM to canted AFM (cAFM) and then to FM[8]. The Chern insulator state has recently been demonstrated in mechanically exfoliated MnBi$_2$Te$_4$ devices at both zero and high magnetic fields[15–18]. However, the topological properties in the cAFM state have not been investigated, where the spin structure can be continuously tuned by a magnetic field. In addition, the electric field effect, which has been demonstrated in 2D magnets[19,20], remains to be explored as a means to control the topological states in thin MnBi$_2$Te$_4$ devices.

Here, we demonstrate that MnBi$_2$Te$_4$ is a Chern insulator in the cAFM state and realize the electric field control of the band topology. We employed a combined approach of polar reflective magnetic circular dichroism (RMCD) measurement to identify the magnetic states and magneto-transport measurement to probe the topological property. To distinguish between electric-field and carrier doping effects, we fabricated MnBi$_2$Te$_4$ devices with dual gates. The devices with a single gate were also used for combined transport and RMCD measurements. The transport and optical measurements were carried out at $T = 50$ mK and 2 K, respectively, unless otherwise specified (see the "Methods" section for fabrication and measurement details).

## Results

**Formation of Chern insulator phase in the cAFM state**. Figure 1a shows the magnetic field dependence of the RMCD signal of a 7-SL MnBi$_2$Te$_4$ device with a single bottom gate (Device 1). Near zero magnetic field, the RMCD signal shows a narrow hysteresis loop due to the uncompensated magnetization in odd layer-number devices[17,21]. Upon increasing the magnetic field, the sample enters the cAFM state at the spin-flop field $\mu_0H_{C1} \sim 3.8$ T, manifested by the sudden jump of the RMCD signal. Remarkably, as the spin–flop transition occurs, the Hall resistivity $\rho_{yx}$ quantizes to about $-h/e^2$ (Fig. 1b), and the longitudinal resistivity $\rho_{xx}$ drops from $\sim$100 kΩ to near zero (Fig. 1c, see Supplementary Fig. 1 for a full gate-dependent transport measurement). As the magnetic field further increases, the canted spins rotate towards the out-of-plane direction and eventually become fully polarized at $\mu_0H_{C2} \sim 7.2$ T with saturated RMCD signal (Fig. 1a). Within the field range of about 3.8–7.2 T, where MnBi$_2$Te$_4$ is in the cAFM state, $\rho_{yx}$ remains quantized with vanishing $\rho_{xx}$, this demonstrates the formation of $C = 1$ cAFM Chern insulator state.

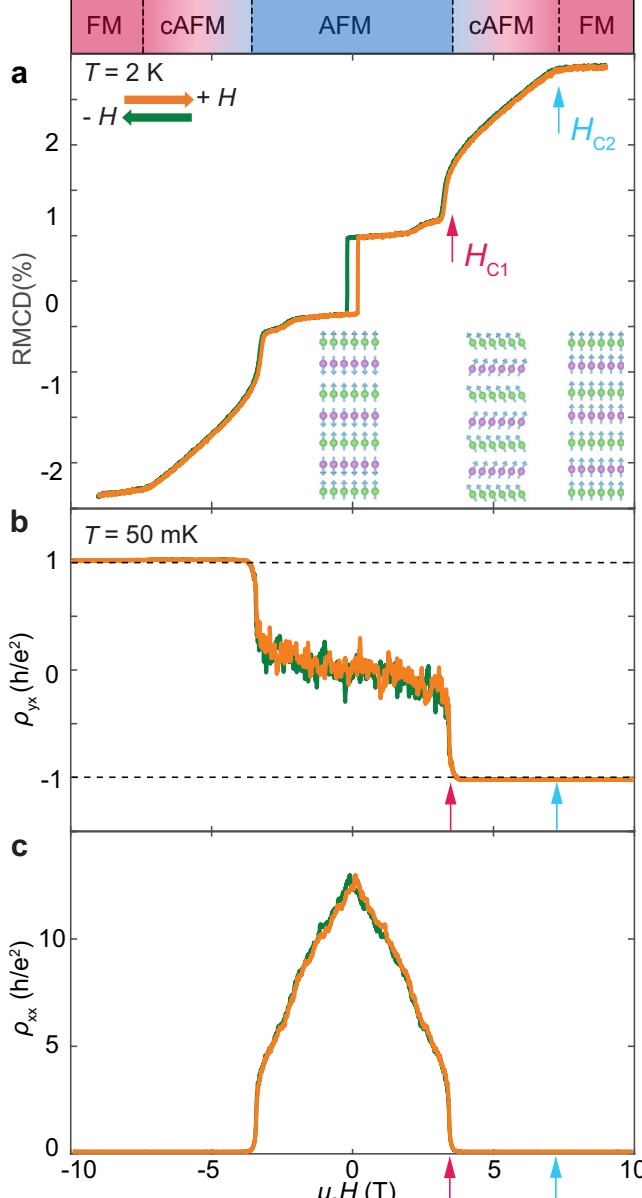

**Fig. 1 Observation of the canted-antiferromagnetic (cAFM) Chern insulator state.** Data is taken from a single gated Device 1. **a** Reflective circular dichroism (RMCD) signal taken at a temperature of 2 K at $V_{bg} = 0$ V. **b** $\rho_{yx}$, and **c**, $\rho_{xx}$ measurements as a function of magnetic field ($\mu_0H$) at optimal gate voltage $V_{bg} = 53$ V. RMCD data are acquired at $T = 2$ K while transport data at $T = 50$ mK. Green and orange traces correspond to magnetic field sweeping down and up, respectively. The red arrow denotes the spin flop field $H_{C1}$ and the light blue arrow indicates the critical field $H_{C2}$ for reaching the field-induced ferromagnetic states. The slight offset of the spin-flop fields between RMCD and transport measurements are due to the different gate voltage and temperature of the measurements. $\rho_{yx}$ is antisymmetrized against $\mu_0H$ while $\rho_{xx}$ is symmetrized.

The cAFM Chern insulator is further supported by exploring the topological phase diagram in dual-gated devices over a broad range of magnetic field. Figure 2a is a 2D color map of $\rho_{yx}$ in a 7-SL dual-gated MnBi$_2$Te$_4$ device (Device 2) as a function of both $\mu_0H$ and gate-induced carrier density $n_G$ at electric field $D/\varepsilon_0 = -0.2$ V/nm (see corresponding $\rho_{xx}$ map and characterization in Supplementary Figs. 2–4). Here $n_G$ partly compensates for the residual carrier

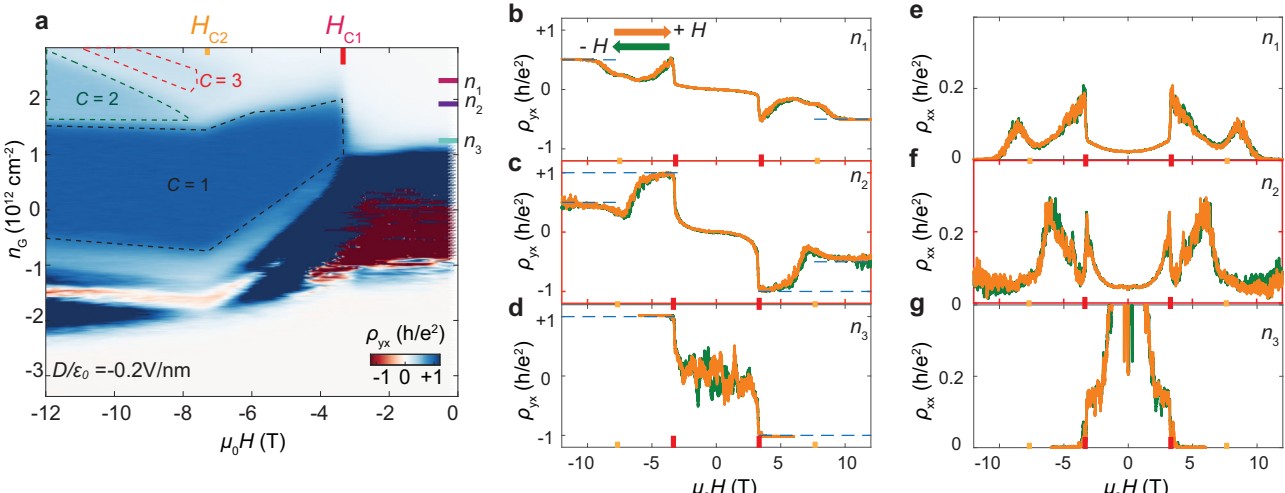

**Fig. 2 Observation of Chern insulator to orbital quantum Hall states phase transition. a** Unsymmetrized $\rho_{yx}$ as a function of magnetic field $\mu_0 H$ and gate induced carrier density $n_G$ at fixed electric field $D/\varepsilon_0 = -0.2$ V/nm in Device 2. Black, green and red dashed lines enclose $C = 1, 2, 3$ quantum Hall states, respectively. The dashed lines are contours defined by $\rho_{yx} = (0.97, 0.47, 0.30)$ $h/e^2$. Carrier densities $n_{1-3}$ and critical magnetic fields $\mu_0 H_{C1}$ and $\mu_0 H_{C2}$ are marked on the axis. $(n_1, n_2, n_3) = (2.23, 1.87, 1.29) \times 10^{12}$ cm$^{-2}$. Note that for $|\mu_0 H| < |\mu_0 H_{C1}|$ and $n_G$ in the range of $-1\text{--}1 \times 10^{12}$ cm$^{-2}$, since the sample is very insulating (i.e. $\rho_{xx} > 1$ G$\Omega$), the excitation current flowing through the sample is not stable and thus $\rho_{yx}$ shows a fluctuating signal with positive and negative values. **b–d** Anti-symmetrized $\rho_{yx}$ as a function of $\mu_0 H$, under different carrier density $n_{1-3}$, and **e–g** The corresponding symmetrized $\rho_{xx}$. Orange and green curves correspond to magnetic field sweeping up and down, respectively.

density in the sample and tunes the Fermi level (see the "Methods" section). The electric field $D$ is defined to be positive when it points from top to bottom gates. Notably, in the range of $n_G$ $1.0\text{--}2.0 \times 10^{12}$ cm$^{-2}$, a sharp phase boundary of the $C = 1$ state is observed near the spin-flop field $|\mu_0 H_{C1}| \sim 3.6$ T. This further supports that the formation of $C = 1$ state is coupled to the cAFM order. The cAFM Chern insulator has been observed in multiple devices. See a device summary in Supplementary Table 1 and their basic characterization in Supplementary Figs. 5–7.

This intimate relationship between the topological and magnetic phase transitions in MnBi$_2$Te$_4$ can be understood as follows. Due to the easy-axis anisotropy, the transition from the AFM to the cAFM state is a first-order phase transition: at the spin–flop field, the magnetization in each SL suddenly rotates into the cAFM state with a finite canting angle. The abrupt change of the magnetic state is accompanied by the formation of the Chern insulator gap, hence a change in the Chern number. Further increase of the magnetic field results in a continuous rotation of the canted spins, which is expected to cause an adiabatic change in the size of the Chern insulator gap until the system enters the FM state. This understanding naturally connects our observation of cAFM Chern insulator and previously reported $C = 1$ Chern insulator in the field–induced FM state[15–17]. As the exchange gap in both cAFM and FM states are adiabatically connected, the stability of Chern number ensures a cAFM Chern insulator state as long as the exchange gap is not closed by external disorder or temperature.

When $\mu_0 H$ is above $|\mu_0 H_{C2}| \sim 7.4$ T, the sample enters the magnetic field-induced FM state. A rich topological phase diagram is uncovered, in which the topological states with corresponding Chern numbers are identified based on $\rho_{yx} \sim h/Ce^2$ and nearly vanishing $\rho_{xx}$. In addition to the $C = 1$, $C = 2$ and 3 states, characterized by $\rho_{yx} \sim h/Ce^2$, appear at higher $n_G$. Unlike the $C = 1$ phase, the contours of $C = 2$ and 3 phases are linearly dependent on magnetic field $\mu_0 H$ (Fig. 2a and its derivative in Supplementary Fig. 2c). This implies that the $C = 2$ and 3 states are a result of the Landau level (LL) formation coexisting with edge state from band topology[22]. For the $C = 1$ phase, there is only a single region present in the phase diagram. So, the $C = 1$

Chern insulator state in the cAFM is adiabatically connected to the same state in the field-induced FM phase, as discussed above. The phase space of $C = 1, 2,$ and 3 states can also be controlled by the top gate, as shown in Supplementary Figs. 3 and 4.

Figure 2b–g plot the $\mu_0 H$ dependence of $\rho_{yx}$ and $\rho_{yx}$ at three selected $n_G$. For heavy electron doping $n_1 \sim 2.23 \times 10^{12}$ cm$^{-2}$, $\rho_{yx} \sim 0$ and $\rho_{xx} \sim 0.005$ $h/e^2$ near zero magnetic field. As $|\mu_0 H|$ increases, we see a kink-like feature, namely a sudden increase of both $\rho_{yx}$ and $\rho_{xx}$ related to the AFM to cAFM transition at $H_{C1}$ (Fig. 2b, e). For $|\mu_0 H| > 10$ T, $\rho_{yx}$ approaches 0.5 $h/e^2$, indicating a $C = 2$ Chern insulator state. For $n_2 \sim 1.87 \times 10^{12}$ cm$^{-2}$, upon entering the cAFM phase at $|\mu_0 H| \sim 3.6$ T, the sample first goes into the $C = 1$ state with $\rho_{yx} \sim h/e^2$ and $\rho_{xx} = 0.05 h/e^2$. At a higher magnetic field $|\mu_0 H| \sim 10$ T, it then switches into the $C = 2$ state with $\rho_{yx} \sim 0.5 h/e^2$ and $\rho_{xx} = 0.05 h/e^2$ (Fig. 2c, f). This phase transition from the $C = 1$ state into a higher Chern number $C = 2$ state as $|\mu_0 H|$ increases at a fixed carrier density is unusual. Increasing $|\mu_0 H|$ increases the degeneracy of LLs. Therefore, if the quantization is caused by the formation of LLs, then the quantum Hall plateau should always change from higher to lower Chern numbers as the magnetic field increases at a fixed carrier density. The opposite observation here further supports our interpretation that the $C = 1$ state observed in the cAFM phase is a Chern insulator originating from the intrinsic nontrivial band structure, while the $C = 2$ state in the FM state is the quantum Hall state due to the formation of LLs[23,24]. For $n_3 \sim 1.43 \times 10^{12}$ cm$^{-2}$, close to the charge neutral point, the transport data shows an abrupt formation of a $C = 1$ Chern insulator at spin-flop field $H_{C1}$, consistent with our discussions above (Fig. 2d, g). The sharp transition exists over a finite doping range, implying that a finite magnetic exchange gap opens suddenly with the spin–flop transition from the AFM to the cAFM phase. This further validates the conclusion that the electronic structure is coupled to the magnetic order[17].

We note that similar effects, i.e., the transition from $C = 1$ to $C = 2$ states as magnetic field increases, have also been studied in other material systems. For example, in doped magnetic topological insulator quantum well (Mn, Hg) Te[23], the increase of magnetic

field first leads to a transition from $C = 2$ to $C = 1$ due to linear Hall effect, then back to $C = 2$ state attributed to the effective nonlinear Zeeman effect of Mn ions. A recent theoretical study[24] on MnBi$_2$Te$_4$ also suggested that the lowest Landau level, stabilized by Anderson localized state, together with quantum anomalous Hall edge state, can form a $C = 2$ state. Thus, the increase of magnetic field can give rise to a transition from $C = 1$ to $C = 2$ state. Note that Weyl semimetal physics has also been proposed in the FM state of bulk MnBi$_2$Te$_4$[25,26]. So, Landau level could originate either from the surface band in a standard magnetic topological insulator picture[15], or from the quantum well states in a confined Weyl semimetal picture[18]. Considering that the 3D bulk crystal of MnBi$_2$Te$_4$ has not been well established experimentally as a Weyl semimetal, the application of the Weyl physics to the atomically thin flakes needs extra caution. Nevertheless, future studies in the FM state are needed to distinguish these two different physical origins of the high Chern number states.

**Electric field control of the cAFM Chern insulator**. The association of the formation of $C = 1$ state with the AFM to cAFM magnetic phase transition suggests the possibility of electric-field control of the Chern number at the cAFM to AFM phase boundary where the magnetic exchange gap should be small. Figure 3a shows $\rho_{yx}$ vs. $\mu_0 H$ at $n_G = 1.0 \times 10^{12}$ cm$^{-2}$ and $D/\varepsilon_0 = -0.3$ V/nm for a dual gated 6-SL MnBi$_2$Te$_4$ device (Device 3), from which we determine the spin–flop field $\mu_0 H_{C1}$ is about 3 T. The small hysteresis loop in the AFM state is possibly due to magnetic domain effects, as previously reported[17]. We then map out $\rho_{yx}$ (Fig. 3b) and $\rho_{xx}$ (Fig. 3c) as a function of $n_G$ and $D$ at

$\mu_0 H_{C1} = 3$ T. The droplet shapes enclosed by the dashed lines in both plots, elongated along the $D$ axis, indicate the $C = 1$ Chern insulator phases. Figure 3d shows both $\rho_{yx}$ and $\rho_{xx}$ as a function of $D$, obtained from line cuts of Fig. 3b and c at $n_G = 1.1 \times 10^{12}$ cm$^{-2}$. As $D$ varies from positive to negative values, the sample starts with small $\rho_{yx}$, enters $C = 1$ state at optimal $D_{opt}/\varepsilon_0 = -0.3$ V/nm supported by the observation of $\rho_{yx} \sim h/e^2$ and vanishing $\rho_{xx}$, and finally exits the $C = 1$ state with reduced $\rho_{yx}$ and increased $\rho_{xx}$. The behavior of $\rho_{yx}$ and $\rho_{xx}$ suggests that the dissipationless chiral edge transport can be switched on and off by an external electric field. The same electric field effect is reproduced in Device 2 (Supplementary Fig. 8).

The sensitive dependence of the $C = 1$ state on $D$ near $H_{C1}$ may be because the electric field directly alters the electronic band structure[27,28], or because it affects the magnetic order (e.g., by tuning $H_{C1}$), which in turn affects the band structure. To distinguish these two mechanisms, we perform gate-dependent RMCD measurements on a 7-SL dual gated device (Device 4). We found that both RMCD signal and spin–flop field are marginally affected by the electric field, but strongly tuned by gate-induced doping (as illustrated in Fig. 4a), consistent with previous reports on 2D magnets[19,20]. Figure 4b shows the RMCD map as a function of $D$ and $n_G$ near spin–flop field $H_{C1}$. As $n_G$ sweeps from electron to hole doping, RMCD significantly increases, i.e., a larger out-of-plane magnetization of the cAFM state at hole doping than electron doping. However, the electric field effect on the magnetic state is marginal, evident by unchanged color along the vertical direction (i.e., parallel to the $D$ axis) in Fig. 4b. The RMCD measurements exclude electric field tuning of the magnetic state as the origin of the electric control of the cAFM Chern insulator state.

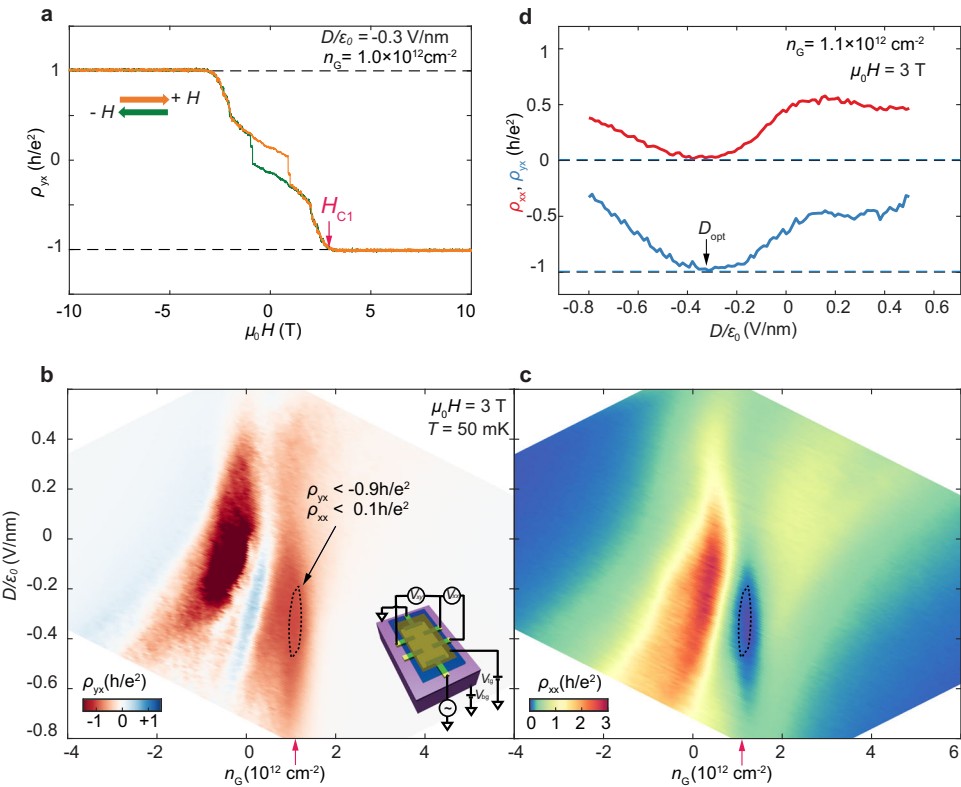

**Fig. 3 Electric control of the cAFM Chern insulator in a dual gated device. a** Anti-symmetrized $\rho_{yx}$ as a function of magnetic field $\mu_0 H$ near optimal doping in Device 3. $\mu_0 H_{C1} \sim 3$ T is identified when $\rho_{yx}$ reaches $-0.999\ h/e^2$. **b, c** $\rho_{yx}$ and $\rho_{xx}$ as functions of gate-induced carrier density $n_G$ and electric field $D$, at fixed magnetic $\mu_0 H = 3$ T. The droplet shapes enclosed by the black dashed line in **b, c** show the quantization area where $\rho_{yx} < -0.9\ h/e^2$ and $\rho_{xx} < 0.1\ h/e^2$. Insert of **b** is a schematic of the dual gated device. **d** $\rho_{yx}$ (blue) and $\rho_{xx}$ (red) vs. $D$ at $n_G = 1.1 \times 10^{12}$ cm$^{-2}$, which are linecuts obtained from **b, c**, indicated by the red arrows. Optimal $D_{opt}/\varepsilon_0 = -0.3$ V/nm denoted by black arrow is identified by $\rho_{yx} \sim h/e^2$ and vanishing $\rho_{xx}$. $\rho_{yx}$ and $\rho_{xx}$ in **b–d** are unsymmetrized.

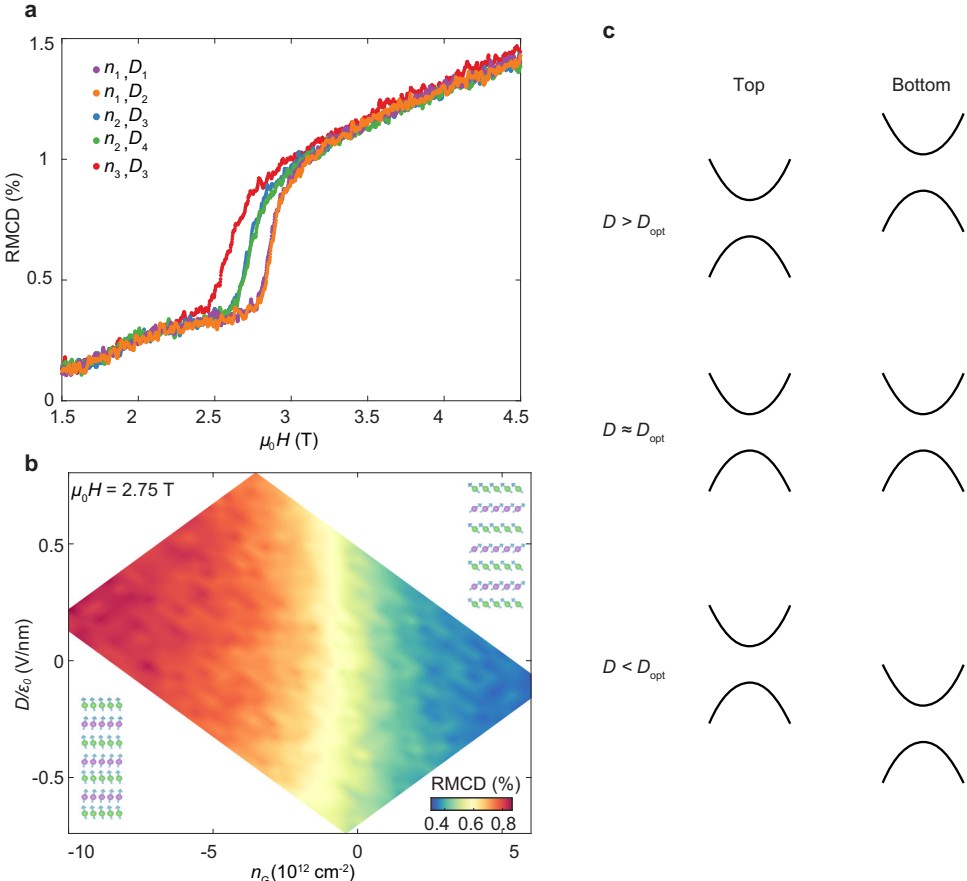

**Fig. 4 Mechanism of electric control of cAFM Chern insulator. a** RMCD as a function of $\mu_0 H$ under selected $n_G$ and $D$. $(n_1, n_2, n_3) = (3.1, -3.1, -9.3) \times 10^{12}$ cm$^{-2}$, $(D_1, D_2, D_3, D_4) = (-0.8, 0.3, 0.8, -0.3)$ V/nm. The data shows that RMCD $(n_1, D_1) =$ RMCD $(n_1, D_2)$, and RMCD $(n_2, D_3) =$ RMCD $(n_2, D_4)$, i.e. for fixed $n_G$, RMCD signal and spin–flop field has neglibigle dependence on $D$. The spin–flop field changes significantly when tuning $n_G$. **b** RMCD map as a function of $D$ and $n_G$ at $\mu_0 H = 2.75$ T. Inset: schematics of cAFM states tuned by $n_G$. The data in (**a**) and (**b**) show that the magnetoelectric effect is solely determined by doping. **c** Illustration of band alignment of top and bottom surfaces under different $D$. See Maintext for details.

## Discussion

This left us with the explanation that the electric field adjusts the relative energy alignment of the magnetic exchange gaps of the two surfaces, as depicted in Fig. 4c. There is a built-in asymmetry between the top and bottom surfaces, possibly due to the different dielectrics used for top and bottom gates. As $D$ is tuned to the optimal electric field ($D_{opt}$), the exchange gaps of top and bottom surfaces are aligned. When the chemical potential is tuned into the exchange gap[29–31], a well-defined edge state with $C = 1$ forms. However, when the deviation of $D$ from $D_{opt}$ is large enough to completely misalign the two magnetic exchange gaps, i.e. the magnetic exchange gap of one surface is aligned to either bulk conduction or valence bands on the other surface, energy dissipative bulk transport is dominant. In the cAFM phase, as the magnetic field increases, the magnetic exchange gap becomes larger as well due to the increase of out-of-plane magnetization. The electric field modulation of the Chern insulator state thus becomes weaker at high field but can be still feasible when the chemical potential is tuned near the band edge (see Supplementary Fig. 9). Our work shows that with the combination of independent control of electric field and carrier doping, as well as intimately coupled magnetic and topological orders, MnBi$_2$Te$_4$ can be a model system for developing on-demand Chern insulator devices and exploring other novel quantum phenomena such as topological magnetoelectric effects.

## Methods

**Device fabrication.** Bulk crystals of MnBi$_2$Te$_4$ were grown out of a Bi–Te flux as previously reported[32]. Scotch-tape exfoliation onto 285 nm-thick SiO$_2$/p-Si substrates was adopted to obtain MnBi$_2$Te$_4$ from 4 SL to 10 SL, distinguished by combined optical contrast, atomic force microscopy, and RMCD measurements. A sharp tip was used to disconnect thin flakes with the surrounding thick flake. Then the flakes were fabricated into single-gated or dual-gated devices. Single-gated devices were fabricated by electron beam lithography with Polymethyl methacrylate (PMMA) resist and followed by thermal evaporation of Cr (5 nm) and Au (50 nm) and liftoff in anhydrous solvents. Then the devices were covered with PMMA as the capping layer. This fabrication process brought spatial charge inhomogeneity and doped the flake. The measured voltages of charge neutrality points shifted between the thermal cycle from 2 to 300 K. Dual-gated devices were fabricated by stencil mask method[33], followed by thermal evaporation of Au (30 nm) and transfer of 30–60 nm h-BN as capping layer by polydimethylsiloxane (PDMS). Standard electron beam lithography (EBL) was adopted to define outer electrodes and metal top gates. Since the highly reflective Au top gate (Device 2 and 3) forbids the RMCD measurements, we also fabricated an optical dual-gated device (Device 4) with graphite top gate (Device 4). Before EBL we transfer a thin graphite (5–20 nm) on top of h-BN with PDMS. For this particular device, a bulk flake which connects to the thin one is used as the contact. In this way, there is no metal underneath or deposited on top of hBN. Thus, Device 4 provided higher doping density than other ones. During the fabrication, MnBi$_2$Te$_4$ was either capped by PMMA/hBN or kept inside an Argon-filled glovebox to avoid surface degradation.

**Transport measurements.** Transport measurements were conducted in a dilution refrigerator (Bluefors) with low-temperature electronic filters and an out-of-plane 13 T superconductor magnet coil. Four-terminal longitudinal resistance $R_{xx}$ and Hall resistance $R_{yx}$ were measured using standard lock-in technique with an a.c excitation of 0.5–10 nA at 13.777 Hz. The a.c. excitation was provided by the SR830 in series with a 100 MΩ resistor, flowed through the device, and was pre-amplified by DL1211 at

1 V/$10^{-6}$ A sensitivity. $R_{xx}$ and $R_{yx}$ signals were pre-amplified by differential-ended mode of SR560 with a 1000 times amplification. All preamplifiers were read out by SR830. A similar amplifier chain provided approximately ±3% uncertainty in the previous study[34]. We estimated our uncertainty was of the same order. In Figs. 1–4, $\rho_{yx}$ overshoot 2.36%, 0.92%, 1.14% and 1.7%, respectively. The sheet resistivity $\rho_{xx}$ and $\rho_{yx}$ were obtained by $\rho_{xx} = s \times R_{xx}/l$ and $\rho_{yx} = R_{yx}$ where $s$ was the width of the current path, $l$ was the length between two voltage probes, estimated from device geometry. Magneto-transport data involved positive and negative magnetic fields was anti-symmetrized/symmetrized by a standard method to avoid geometric mixing of $\rho_{xx}$ and $\rho_{yx}$. As this mixing did not affect the topological transport signal, we presented raw data for fixed magnetic field study. To convert top gate voltage $V_{tg}$ and bottom gate voltage $V_{bg}$ into gate-induced carrier density $n_G$ and electric field $D$, we used $n_G = (V_{tg}C_{tg} + V_{bg}C_{bg})/e$ and $D/\varepsilon_0 = (V_{tg}C_{tg} - V_{bg}C_{bg})/2\varepsilon_0$, where $C_{tg}$ and $C_{bg}$ are top and bottom gate capacitance obtained from device geometry, $e$ the electron charge and $\varepsilon_0$ the vacuum permittivity. This formula is derived from the parallel-plate capacitor model. Fixing $D(n_G)$ and sweeping $n_G(D)$ monotonically modify the carrier density $n_{2D}$ (external displacement field $D_{ext}$) and thus the chemical potential (electric field) of the device. To obtain the $n_G$-$D$ maps, we first got $V_{tg} - V_{bg}$ maps of transport data by sweeping $V_{bg}$ from the negative side to the positive for every fixed $V_{tg}$. Converting $V_{tg} - V_{bg}$ to $n_G$-$D$ leads to the uncovered parameter space, e.g., in Figs. 3b, c and 4b. The $n$-$\mu_0 H$ ($V_{bg}$-$\mu_0 H$) maps were taken by sweeping dual gates (back gate) quickly, ~4 min per data line back and forth, while sweeping magnetic field slowly, ~0.015 T/min. This method gave digitized noise or unfinished lines near the field limit, e.g., near 0 T of Fig. 2a.

**Reflective magnetic circular dichroism measurements.** The experiment setup follows our previous RMCD/MOKE study of magnetic order in CrI₃. RMCD measurements were performed in an attoDRY cryostat with attocube $xyz$ piezo stage, the base temperature of 1.6 K and 9 T superconducting magnet. The magnetic field was applied perpendicular to the sample plane. Linearly polarized 632.8 nm He–Ne laser with 200 nW power was focused through an aspheric lens to form ~2 μm beam spot on the sample surface. The out-of-plane magnetization of the sample induced magnetic circular dichroism (MCD) $\Delta R$, the amplitude difference between the reflected right- and left-circularly polarized light. To obtain the RMCD $\Delta R/R$ signal, two lock-in amplifiers SR830 were used to analyze the output signals from a photomultiplier tube with the chopping frequency $p = 1.377$ kHz and photoelastic modulator frequency $f = 50$ kHz. The ratio between $p$-component signal $I_1$ and $f$-component signal $I_2$ is proportional to the RMCD signal: $\Delta R/R = I_2/(J_1(\pi/2) \times I_1)$ where $J_1$ is the first-order Bessel function.

## Data availability

Source data of Figs. 1–4 can be found at: https://doi.org/10.6084/m9.figshare.19193498.v4. All other data that support the findings of this study are available from the corresponding author upon reasonable request.

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

## Acknowledgements

The authors thank Chaoxing Liu for insightful discussions. The electrical control of Chern number in the canted antiferromagnetic states was mainly supported by AFOSR FA9550-21-1-0177. Magneto-optical measurements and theory understanding were supported as part of Programmable Quantum Materials, an Energy Frontier Research Center funded by the U.S. Department of Energy (DOE), Office of Science, Basic Energy Sciences (BES), under award DE-SC0019443. The authors also acknowledge the use of the facilities and instrumentation supported by NSF MRSEC DMR-1719797. J.Y. acknowledges support from the U.S. Department of Energy, Office of Science, Basic Energy Sciences, Materials Sciences and Engineering Division. C.-Z.C. acknowledges the partial support from the Gordon and Betty Moore Foundation's EPiQS Initiative (Grant GBMF9063). Y.-T.C. acknowledge support from NSF under award DMR-2004701, and the Hellman Fellowship award. X.X. and J.-H.C. acknowledge the support from the State of Washington funded Clean Energy Institute.

## Author contributions

J.C. and D.O. fabricated the devices, assisted by Z.F., M.H., and Z.L. J.C. and D.O. performed the transport measurements, assisted by Z.F. J.C., Z.L. and T.S. performed the RMCD measurements. C.W. and D.X. provided theoretical support. J.Y. synthesized and characterized MnBi₂Te₄ bulk crystals. X.X., J.Y., D.X., C.-Z.C, Y.-T.C. J.-H.C. D.C. supervised the project. J.C., D.X., C.-Z.C., X.X., Y.-T.C, and D.O. wrote the manuscript with inputs from all authors. All authors discussed the results.

## Competing interests

The authors declare no competing interests.
