## [Peer Review File · Nature Communications]

REVIEWER COMMENTS

Reviewer #1 (Remarks to the Author):

The manuscript by J.Cai et al. describes coexistence of magnetism and topology in MnBi₂Te₄. The idea is to observe the signatures of canted-antiferromagnetic (cAFM) Chern insulator and its signatures/evolution in electric fields. I think the paper is quite interesting; however, the understanding of the results is questionable. In Fig.1, the (cAFM) Chern insulator can be seen only as an extra kink in the reflective circular dichroism (RCD) signal while transport properties do not allow distinguishing (cAFM) Chern insulator from usual Chern insulator with all layers fully polarized. However, in Fig.2, only transport data are presented for dual-gated devices and not RCD signal. I am guessing that in dual-gated devices RCD signal cannot be measured easily. Why do the transport features prove an existence of (cAFM) Chern insulator for dual gate configuration but not for a single gate? I feel that further experimental and theoretical evidence needs to be given to prove cAFM Chern insulator phase-

Usual theoretical picture for MnBi₂Te₄ is related to Dirac surfaces under influence of an exchange field and an out-of-plane magnetic field. This explains for example all plateaus, which are forming in Hall resistance and are electrically tuned by gate in Science 367, 895 (2020) (Ref. 15 in the current paper). To explain the data in the manuscript, one needs to add to this picture systematically the magnetic layer anisotropy and possibly the electron-electron interaction between the layers. This should allow unambiguously understand Hall resistance in Fig.2b. This should be done to believe in the interpretation of the experimental results.

Summarizing, I believe that the manuscript in the current form is not clear or convincing enough to be published in Nature Communications. I will be willing to reconsider, after the authors significantly improve presentation and give extra experimental and theoretical arguments for the existence of cAFM Chern insulator along the lines I described above.

Reviewer #2 (Remarks to the Author):

Dear Authors,

In this manuscript, the authors demonstrate the electrical control of Chern insulator states in MnBi₂Te₄ devices. By applying optical technique under magnetic field, the magnetic phase transition was revealed from antiferromagnet (AFM) at zero magnetic field to canted AFM and FM (Fig. 1) at high field. Judging from Fig. 1, the exchange gap is formed at canted AFM and FM above H_{c1}. Quantized anomalous Hall state appears with C = 1 when Fermi energy is controlled at both the exchange gap at top and bottom surface states. Under the specific condition with large nG, C = 2 state appears in Fig. 2a, which originates from Landau level formation. The observation of high Chern number is interesting.

It would be great that following comments are helpful to improve the manuscript.

1, Why four devices are discussed in this study?

Reproducibility is fine. Device 1 is single gate and a rest of three is dual gate. RCD can be applied to both single (Fig. 1a) and dual (Fig. 4a) gate devices. Why did not the authors simultaneously measure transport and MCD with the identical device (Device 2 or 3)?

In Device 4, the charge carrier density is rather large compared to the condition discussed in Fig. 1-3. It would be better to explain this point in main text.

In Device 3 with 6 SL, why the residual large hysteresis is shown in r_{xy} at zero magnetic field in Fig. 3a? Are the results shown in Fig. 3b-d really significant? In ref. 2 for the first experimental observation of QAHE in Cr-doped topological insulator, the control of C = 1 or

other has been demonstrated. In other words, the Fermi energy position can be controlled in or far from the exchange gap position. In page 4, there is a description [similar results are also observed with Device 2 in supplement]. Are there different merits in the comparison with four devices? If it is there, it would be great to emphasize this point in main text.

It would be great that the photos of all devices are shown in supplement. Then, please compare the sample size and spot size of laser for MCD measurement. In addition, please describe clearly that the reason why MCD can be measured on dual gate device. Probably, the top electrode is transparent to the wavelength of light. For general readers, please explain this point clearly.

2, Please address the mobility of charge carrier in the device. In addition, please define which is electron $+nG$ or $-nG$. For the discussion about the feasibility of LL formation, cyclotron gap size or broadening of the LL state are important parameters. If the authors measure the ordinary Hall effect above Neel temperature, the values of charge carrier density and mobility are quite informative. If not, by applying the estimate values of nG and the measured sheet resistance, the mobility can be discussed. The feasibility of the appearance of $C = 2$ may be also examined with the relation between the value of nG and magnetic field in simple formula based on density of states of linear bands. It would be great that these points are explained in main text.

3, In Fig. 2a, color code may be misleading because the red region looks $C = -1$ state or larger than $-1 h/e^2$ and dark blue region around low magnetic field looks larger than $+1 h/e^2$. However, no experimental raw trace data is provided in main text and supplement. The readers cannot recognize what happens at low magnetic field and low nG region. It would be great that the raw trace data of r_{yx} and r_{xx} for $nG = +0.5 \times 10^{12}$ and $-0.5 \times 10^{12} \text{ cm}^{-2}$ are shown in supplement. It is rather difficult to distinguish the values of r_{xx} and r_{yx} in the color map. Then, the authors explain what happens around there. For general readers, the reason why insulator without surface state at antiferromagnetic condition at zero magnetic field should be also explained.

4, As for the appearance of $C = 2$ at high magnetic field and large nG in Fig. 2, probably two prerequisites should be fulfilled. One is location of Fermi energy. For example, at n_2 in Fig. 2c, did the exchange gap position shift against Fermi energy by applying magnetic field? Fermi energy should locate far from both the exchange gap at top and bottom surface states to appear $C = 2$, because if Fermi energy locates in the exchange gap at either surface, Chern number can only be 1 with LL formation at other surface. It seems reasonable that the shift of Fermi energy by applying electric field or dual gating. However, around n_2 in Fig. 2, the transition occurs from $C = 1$ to $C = 2$ by increasing magnetic field, which is one of main achievement in this study. It would be great that the authors clearly explain the mechanism for this transition.

In addition, the claim for the observation of $C = 3$ seems too strong because plateau in r_{yx} is narrow and difficult to recognize the accuracy of quantized value. The magnified result of r_{yx} and r_{xx} shown in supplement is helpful to understand it.

The asymmetry between electron and hole region may not be explained clearly in the present manuscript.

At zero magnetic field, the basic transport property should be shown in supplement such as r_{xx} as a function of nG for all devices, because color map does not work to compare the values. The comparison with line data will evidence the reproducibility in this study. If the value of r_{xx} is rather high, of course the r_{xx} as a function of nG at high field is alternatives.

5, H_{c2} looks shifted in n_2 in Fig. 2f from in n_1 in Fig. 2e while H_{c1} is kept constant. The shift of H_{c2} can be detected by MCD? This shift may be related on the appearance of $C = 2$ at high magnetic field. While the H_{c1} variation with applying electric field is discussed in

Fig. 4, the presented result is limited magnetic field region. The description of [The RMCD measurements exclude electric field tuning of the magnetic state as the origin of the electric control of the Chern insulator state.] may be not clear. Because the transition from $C = 1$ to $C = 0$ or 2 is discussed in this manuscript, the consistent discussion is necessary through overall in this manuscript.

6, is the description of [Increasing $\mu_0 H$ increases the degeneracy of LLs] in page 3 accurate? Could the authors recognize that the $C = 1$ comes from LL formation or QAH exchange gap? This point seems not clearly discussed in the present manuscript. It would be great that the authors define the region for QAHE-driven $C = 1$ or LL-driven $C = 1$ if it exists.

7, Can the authors comment on the controllable energy range based on Fig. 4c? In the devices in this study, large value of charge carrier density is controlled roughly 1×10^{13} in Fig. 4. By applying dual gating, how large energy range can be controlled in Fig. 4c? Controllable range is much larger than the gap size? Could the authors control only Fermi energy with keeping $D = D_{opt}$ condition? Naively, to observe $C = 2$, Fermi energy locates far from both gap at top and bottom. It would be great that the authors address the controllable range of energy in the device. Then, it is further great that the comparison between the controllable range and gap size rough quantitatively.

8, In Discussion part, the description of [the magnetic exchange gap increases. The electric field modulation of the Chern insulator state becomes weaker but is still feasible.] is not well organized. The origin for the increase of the magnetic exchange gap seem comes from the enhancement of exchange coupling owing to increase of out-of-plane magnetization. If so, the gap size is saturated at H_{c2} at ferromagnetic condition. This point should be described.

The reason why the electric field modulation is weakened is not clearly discussed in anywhere in the present manuscript. It would be great that these descriptions are revised with explanation based on the experimental results.

Technical comment

1, It is rather difficult to recognize the yellow line and region indicating $C = 3$ in Fig. 2a. Please change color.

2, The color code of Fig. 3c seems confusing because blue region is widely shown. However, most of the blue region in r_{xx} corresponds to zero in r_{yx} in Fig. 3b. If so, it would be great that the addition of periodic contour line roughly every h/e^2 may be helpful to recognize. Please improve the color.

3, In method section, device fabrication, please add a reference at [as previously reported].

Reviewer #3 (Remarks to the Author):

In this manuscript, Cai et al systematically studied the electrical transport behaviors of the recently discovered intrinsic AFM TI $MnBi_2Te_4$. They found the Chern insulator state with quantized R_{yx} and zero R_{xx} appears at a low magnetic field of 3.5 T corresponding to a canted AFM state identified by RMCD measurements. By applying a magnetic field, the Chern insulator with $C = 1$ is driven into orbital quantum Hall states but with higher Chern numbers. In addition, the authors fabricated a dual-gated device to demonstrate the effects of electric field and carrier doping on manipulating the topological and magnetic properties of $MnBi_2Te_4$. This work stands in the continuation of efforts on the studies of the first van der Waals topological magnet $MnBi_2Te_4$ that has been proposed and demonstrated to host

a rich variety of topological quantum phases, such as the Chern insulator and the axion insulator. Although there have already been relevant reports on the field-induced Chern insulator state in MnBi₂Te₄, the Hall quantization in most previous works is attributed to the formation of FM rather than canted AFM as the authors found in the experiment. The C = 1 to C = 2 phase transition indicates both the Chern insulator (quantum anomalous Hall effect) and quantum Hall effect can coexist in MnBi₂Te₄, which is absent in magnetically doped TIs.

However, I find the claim of the canted AFM Chern insulator state unconvincing in its line of reasoning. More reproducible results are required to pin down whether this claim arises from accidental observation. Of particular note is that there are inconsistencies between the current data and the results in early works, which are never discussed in the manuscript. The discussion on the physical origins is also too simple. Importantly, I find the surface exchange gap picture can even give rise to contradictory expectations to the experiments. For the C = 1 Chern insulator to C = 2 transition, the authors claimed the former one is due to the exchange gap and the latter is due to Landau levels. However, a Chern insulator state should have its Fermi level exactly located in the gap. In the quantum Hall state, the Fermi level should be outside the bandgap (in the conduction band for the present situation) so that the Landau levels play a role. Generally, these two topological states exist exclusively and cannot be switched between each other simply by applying a magnetic field. The theoretical discussion should be modified.

In addition to the magnetic field tuned topological phase transition, the role of the electric field in adjusting the relative energies of the exchange gaps is also questionable. I make the statement based on the phase diagram in supplementary Fig. 2. Tuning the relative energies of the exchange gap will shrink the window of V_g for observing the C = 1 phase. However, judging from the experimental data, not only the shape but also the area of the C = 1 phase is insensitive to V_g. To put it in the least way, regardless of whether the picture can account for the data, whether there are surface states in FM MnBi₂Te₄ remains a question. There is no doubt AFM MnBi₂Te₄ is a TI. But for its FM state, many theories have pointed out switching magnetic structure from AFM to FM can lead to a Weyl semimetal (PRL 107, 127205 (2011), PRL 122, 206401 (2019) and Sci. Adv. 5, eaaw5685 (2019)). In this situation, the concept of surface state, as well as the exchange gap, no longer exists. Several experimental works also provide evidence for the field-induced Weyl semimetal state, as shown in the two early works (arXiv:2001.08401 and arXiv:2002.10683). Interestingly, the Weyl semimetal picture gives a more comprehensive explanation of the presence of quantum Hall state in MnBi₂Te₄ Chern insulator.

In spite of all these concerns, I still find the topic in the manuscript is actual, and the reported experimental phenomena are interesting enough. I believe it would draw extensive attention from researchers in the fields of topological state of matter and material science. However, at the current stage, I cannot recommend this article without significant revisions. I suggest the authors provide more supportive data and give more reasonable explanations to their observations.

Please consider the following questions and comments:

1. I find there are some inconsistencies with the RMCD data for the three 7-SL devices. The authors should explain why samples with the same thickness have different H_{c1}.
2. The authors compared the RMCD signals with the transport data to identify the C = 1 Chern insulator phase is in canted AFM state (Fig. 1). Unfortunately, this comparison is inaccurate. Because both the transport and magnetic properties of MnBi₂Te₄ are temperature dependent. A reasonable comparison should be made at the same

temperature, such as at 2 K.

3. "Chern insulator phase in the canted AFM state" is an important claim. However, I find this conclusion is drawn mainly from the results of Device 1 (Fig. 1). Considering the large discrepancies of H_{c1} in the samples and the lack of comparisons between RMCD and transport data in Devices 2 and 4, I cannot rule out the possibility that this conclusion arises from cherry-picked data. I make the statement also based on comparing Fig. 1 with previously published results (Science 367, 895 (2020), Nat. Mater. 19, 522 (2020), Nat. Sci. Rev. 7, 1280 (2020), Nano Lett. 21, 2544 (2021)). In these reports, most groups found the Chern insulator state appears in the FM state rather than in the canted AFM state. Even in the Nano Letter work by some of the authors, the samples do not show quantitation until entering the FM state. Although I believe this finding is interesting, more reproducible data are required to convince general readers. What about the transport data for Device 4? How many samples are measured and does the Chern insulator state always appears in the canted AFM state?

4. "Unlike the $C = 1$ phase, the contours of $C = 2$ and 3 phases are linearly dependent on magnetic field μ_0H (Fig. 2a). This implies that the $C = 2$ and 3 states are a result of the Landau level (LL) formation coexisting with edge state from band topology (ref. 21)". The linearity of the $C = 2$ and $C = 3$ phases is only marked by the manually added dashed lines. I suggest the authors show the first derivative data. In fact, the $C = 3$ phase in Fig. 2a is not clear. Even in Figs. 2b and 2c, it is difficult to identify the $\rho_{yx} = h/3e^2$ and $\rho_{yx} = 0$ quantum Hall plateau. It seems the claim of $C = 3$ quantum Hall state is an overstatement. In addition, could the authors estimate the sample mobility so that readers may judge the relevance of the Landau levels to the high Chern number state?

5. As I have mentioned at the beginning, one major drawback of the theoretical picture (surface exchange gap + Landau levels) is that it only provides a possible explanation to $C = 2$ quantum Hall state but cannot explain why it could arise from a $C = 1$ Chern insulator. For fixed Fermi level position, depending on in the exchange gap or in the conduction band, the system should be either a Chern insulator or a quantum Hall state (if Landau levels form).

6. Regarding the opinion "the Landau level (LL) formation coexisting with edge state from band topology", it has already been demonstrated experimentally by transport studies of $MnBi_2Te_4$ flakes (arXiv:2001.08401). Differently, in this arXiv work, the Chern insulator to quantum Hall state transition is explained on the basis of Weyl semimetal picture rather than magnetic TI. In my eyes, depending on the number of Landau levels in the conduction subband, the Weyl model can naturally explain the $C = 1$ Chern insulator to $C = 2$ quantum Hall phase transition observed in the present manuscript.

7. "The behavior of ρ_{yx} and ρ_{xx} suggests that the $C = 1$ state can be switched on and off by an external electric field". This statement is inaccurate. The experiment was carried near the phase boundary. A lot of factors can easily smear out the Chern insulator quantization. The deviation of ρ_{yx} from h/e^2 to an unquantized value and the appearance of ρ_{xx} do not mean the Chern insulator is turned off but only indicate the presence of diffusive conduction channels. A real "switch on and off" refers to topological phase transition with gap close and reopen. Therefore, the "electric field control of the band topology" is not realized.

We sincerely thank all three referees for their thoughtful and constructive comments. In the following, we provide a point-to-point response to their comments. Major changes are shown in red in the revised manuscript.

Response to Referee 1

The manuscript by J.Cai et al. describes coexistence of magnetism and topology in MnBi₂Te₄. The idea is to observe the signatures of canted-antiferromagnetic (cAFM) Chern insulator and its signatures/evolution in electric fields. I think the paper is quite interesting; however, the understanding of the results is questionable.

We thank the reviewer for considering our work quite interesting. We hope our responses will address his/her concerns below.

1. In Fig.1, the (cAFM) Chern insulator can be seen only as an extra kink in the reflective circular dichroism (RMCD) signal while transport properties do not allow distinguishing (cAFM) Chern insulator from usual Chern insulator with all layers fully polarized. However, in Fig.2 , only transport data are presented for dual-gated devices and not RCDM signal. I am guessing that in dual-gated devices RCDM signal cannot be measure easily. Why does the transport features prove an existence of (cAFM) Chern insulator for dual gate configuration but not for a single gate?

Response: We respectfully disagree with Referee 1 that the appearance of the cAFM Chern insulator state is not verified in our single gated device. The combined transport and RMCD results shown in Fig.1 demonstrate the presence of the cAFM Chern insulator state. Specifically, the RMCD data is used to confirm the cAFM state in our MnBi₂Te₄ device, while the transport data is used to verify the appearance of the quantized Hall resistance and vanishing longitudinal resistance in this cAFM regime.

Fig. R1 shows ρ_{xx} and ρ_{yx} as a function of backgate and magnetic field for Device 1 with a single gate, in good agreement with the RMCD data in the maintext. This phase diagram reveals a similar $C = 1$ cAFM state as that in Device 2 with dual gates in Fig. 2a. Since the dual gated device can enable independent control of displacement field and carrier density, this capability enables us to achieve a full phase diagram,

Figure R1. Gate dependent transport of Device 1. **a-b**, ρ_{xx} (**a**) and ρ_{yx} (**b**) as a function of $\mu_0 H$ and $V_{bg} - V_{bg}^0$. **c**, RMCD adopted from Fig. 1 in the Maintext. The dashed lines indicate H_{c1} and H_{c2}

such as the observation of the $C = 2$ and $C = 3$ states (Fig. 2a) and electric field control of the cAFM Chern insulator state near the cAFM-AFM phase transition boundary.

For the benefit of any reader who has the same concern as the reviewer, we have added **Figure R1 in our Supplementary Materials as Supplementary Fig. 1, and the relevant discussion in the revised manuscript.**

2. I feel that further experimental and theoretical evidence needs to be given to prove cAFM Chern insulator phase- Usual theoretical picture for MnBi₂Te₄ is related to Dirac surfaces under influence of an exchange field and an out-of-plane magnetic field. This explains for example all plateaus, which are forming in Hall resistance and are electrically tuned by gate in Science 367, 895 (2020) (Ref. 15 in the current paper). To explain the data in the manuscript, one needs to add to this picture systematically the magnetic layer anisotropy and possibly the electron-electron interaction between the layers. This should allow unambiguously understand Hall resistance in Fig.2b. This should be done to believe in the interpretation of the experimental results.

Response: From the experimental point of view, we believe that our data unambiguously established the cAFM Chern insulator. (1) We employed RMCD to determine the magnetic states. (2) We performed electrical transport measurement to demonstrate the topological invariant. Our data shows that as cAFM state forms, ρ_{yx} is quantized while ρ_{xx} approaches zero, confirming the $C = 1$ Chern insulator state in the cAFM state.

We thank the referee for pointing out possible theoretical pathways to understand the cAFM Chern insulator state formation. We would like to point out that since the lone single device presented in Ref. 15 shows the zero-field QAH state, many groups have been trying hard to reproduce this result. Unfortunately, it has yet to be succeeded. On the other hand, the Chern insulator state with $C = 1$ in the field-induced FM state has been observed by many groups (Refs. 15-18). Therefore, the FM Chern insulator state should be robust. Let's start from this state. Decreasing magnetic field introduces an FM to cAFM magnetic phase transition. This magnetic phase transition is of second-order type, which results in a continuous reduction of the magnetic exchange gap. During this process, the Chern number should remain one, as long as the gap size is larger than the thermal excitation energy, which we demonstrate in the present work.

To understand the $C = 1$ to $C = 2$ transition in Fig. 2, we note that there are indeed ongoing theoretical efforts such as PRL127, 236402 (2021) (published on December 3, 2021, posted in arXiv (arXiv:2107.04752) after the submission of our paper), which we now added as a reference in the revised manuscript. This PRL paper proposes that for MnBi₂Te₄ under high magnetic field, the lowest Landau level, formed by Anderson localized state, together with QAH edge state, leads to a $C = 2$ state. Similar effects have also been studied in other material systems. For example, PRB 102, 165305 (2020) observed a $C = 2$ to $C = 1$ and then back to $C = 2$ transition, attributed to two species of holes in the system. In PRB 85, 195304 (2012), in a related magnetically doped quantum well (Mn, Hg)Te, a reentrant quantum Hall effect (transition from $C = 2$ to $C = 1$ due to linear Hall effect, and then enters $C = 2$ again as magnetic field increases) is predicted due to Mn ions' effective nonlinear Zeeman effect. Here, the observed transition from $C = 1$ to $C = 2$ as the magnetic field increases in MnBi₂Te₄ could share the same origin as this quantum well system.

In the revised manuscript, we have added the above discussion and cited PRB 85, 195304(2020) and PRL127, 236402 (2021).

Response to Referee 2

In this manuscript, the authors demonstrate the electrical control of Chern insulator states in MnBi₂Te₄ devices. By applying optical technique under magnetic field, the magnetic phase transition was revealed from antiferromagnet (AFM) at zero magnetic field to canted AFM and FM (Fig. 1) at high field. Judging from Fig. 1, the exchange gap is formed at canted AFM and FM above H_{c1}. Quantized anomalous Hall state appear with C = 1 when Fermi energy is controlled at both the exchange gap at top and bottom surface states. Under the specific condition with large nG, C = 2 state appears in Fig. 2a, which originates from Landau level formation. The observation of high Chern number is interesting.

We appreciate that the reviewer finds our result interesting.

It would be great that following comments are helpful to improve the manuscript. 1, Why four device are discussed in this study? Reproducibility is fine. Device1 is single gate and a rest of three is dual gate. RMCD can be applied to both single (Fig. 1a) and dual (Fig. 4a) gate devices. Why did not the authors simultaneously measure transport and MCD with the identical device (Device 2 or 3)?

Response: There are a couple of motivations to present multiple devices. (1) It is our true belief that reproducibility is necessary to establish any conclusion. In light of Reviewer #3's request of further confirmation of reproducibility, we have added an additional device (Device 5) in the revised Supplementary Materials (See Supplementary Fig. 7) to confirm the C = 1 cAFM Chern insulator state. (2) For dual gated transport device, there is an ~40nm thick Au metal top gate, which prevents the optical (*i.e.*, RMCD) measurements. One solution is to use a graphite top gate. However, the shaping of graphite top gate requires an etching process that degrades MBT samples because MBT is highly air- and temperature-sensitive. This makes it challenging to fabricate a dual gated device for simultaneous transport and RMCD measurements.

The collective evidence of all devices supports our conclusion. For the reviewer's convenience, we summarize the types of devices and results below.

- i) Single gated devices allow RMCD and transport obtained in the same devices. (Device 1 in main text, and Device 5 in Supplementary Materials) Using these devices, we show that as cAFM state is formed, quantized ρ_{yx} and vanishing ρ_{xx} are observed. This demonstrates the formation of cAFM Chern insulate state.
- ii) Dual gated device with metal top gate, where transport with individual control of carrier doping and electrical field is done (Device 2 and 3). These dual-gate devices show electrical control of the cAFM state near the phase transition boundary.
- iii) Dual gated device with a graphite top gate, where we identify that doping dominates the modulation of the magnetic states, but not the electrical field (Device 4). This leads us to

propose a physical picture in understanding the electrical field control of the cAFM state near the AFM-cAFM phase boundary in the dual gated device.

In the revised manuscript, we added more clarifications about why we used multiple devices in Methods section of the revised manuscript.

In Device 4, the charge carrier density is rather large compared to the condition discussed in Fig. 1-3. It would be better to explain this point in main text.

Response: For Device 1 in Fig. 1, its carrier density is limited because it has only one gate (SiO₂/p-doped silicon backgate).

For Devices 2 and 3 in Figs. 2-3, a top gate is added to tune the carrier density within a broad range. However, the metallization of the Au top gate causes degradation of the hBN dielectric layer, which results in a lower leaking gate voltage than that of a pristine hBN (Fig. R2).

For Device 4 in Fig. 4, the transferred graphite reduces the degradation of the hBN, and thus the leaking gate voltage is higher than Devices with Au top gate. Note that we used a piece of bulk MnBi₂Te₄ which is connected to the thin one as the contact, and thus there is no metal part underneath hBN (Fig. R2). These combined factors allow us to achieve a higher carrier density in Device 4.

We added the above information in Methods section of the revised manuscript.

In Device 3 with 6 SL, why the residual large hysteresis is shown in r_{yx} at zero magnetic field in Fig. 3a?

Response: We have studied this particular effect in our previous reports on 6SL transport (Ref. 17, Nano Lett. 21, 2544-2550 (2021)). We found that the small hysteresis loop is likely due to the magnetic domain effects near the edge of the device.

We have briefly discussed the possibility of the observed hysteresis loop in our 6 SL devices in the revised manuscript.

Are the results shown in Fig. 3b-d really significant? In ref. 2 for the first experimental observation of QAHE in Cr-doped topological insulator, the control of $C = 1$ or other has been demonstrated. In other words, the Fermi energy position can be controlled in or far from the exchange gap position.

Response: The QAHE in Cr-doped topological insulator thin films, first discovered by one of the co-authors in this manuscript, indeed shows single gate control of the $C = 1$ QAH state. However, a single gate was used to tune only the carrier density of the sample. In other words, this single gate is used to tune the chemical potential into and out of the magnetic exchange gap, which corresponds to the QAH state with chiral edge transport and the bulk conduction transport. We note that this single gate control in the original QAH work does not alter the band structure of the magnetically doped TI film, which is physically different from our electric field control of the relative alignment of the magnetic exchange gaps between top and bottom surfaces at the fixed carrier density.

In page 4, there is a description [similar results are also observed with Device 2 in supplement]. Are there different merits in the comparison with four devices? If it is there, it would be great to emphasize this point in main text.

Response: This question is similar to the previous comment, “Why four devices are discussed in this study”. We hope that our above response has clarified why we studied multiple devices in this work.

It would be great that the photos of all devices are shown in supplement. Then, please compare the sample size and spot size of laser for MCD measurement.

Response: The photos of all our devices are included in Fig. R2, with the sample size indicated by the scale bar. The laser spot size is about $1 \mu\text{m}$ (see Methods section).

We added Fig. R2 as Supplementary Fig. 6 in the revised Supplementary Materials.

In addition, please describe clearly that the reason why MCD can be measured on dual gate device. Probably, the top electrode is transparent to the wavelength of light. For general readers, please explain this point clearly.

Response: As noted above, the metal top gate in Devices 2 and 3 prevented the RMCD measurements. Device 4 uses thin graphite as the top gate, which is transparent for our RMCD measurements.

2, Please address the mobility of charge carrier in the device. In addition, please define which is electron $+nG$ or $-nG$. For the discussion about the feasibility of LL formation, cyclotron gap size or broadening of the LL state are important parameters. If the authors measure the ordinary Hall effect above Neel temperature, the values of charge carrier density and mobility are quite

informative. If not, by applying the estimate values of n_G and the measured sheet resistance, the mobility can be discussed

Response: We thank the referee for bringing up the mobility issue. We now add the mobility information of the devices used in our transport measurements in **Table R1**. To obtain mobility, we dope the device far away from the anomalous Hall region. We then extract Hall carrier density and mobility by fitting the linear Hall effect with respect to a small magnetic field.

We added **Table R1** as **Supplementary Table. 1** in the revised **Supplementary Materials**.

Device type	Device # in the text	Thickness (# of SLs)	cAFM Chern insulator?	Mobility ($10^3 \text{cm}^2 \text{V}^{-1} \text{s}^{-1}$) (50mK, $\sim 3 \times 10^{12} \text{cm}^{-2}$)
Single gated device	1	7	Y	0.2
	5	8	Y	0.9
Dual gated device	2	7	Y	1.7-3
	3	6	Y	1.5-2

Table R1. A list of the transport devices. To obtain electron, we gate the device to be far away from anomalous Hall region and extract Hall carrier density and mobility by fitting the linear Hall effect with respect to small magnetic field.

The feasibility of the appearance of $C = 2$ may be also examined with the relation between the value of n_G and magnetic field in simple formula based on density of states of linear bands. It would be great that these points are explained in main text.

Response: Note that in a previous study, Ref. 15 found that only a small portion of electrons contributes to Landau levels. So, the fan diagram can be represented by $n_L = P(n_G - n_0) = \frac{B\nu}{\phi_0}$, where n_0 is the carrier density that fills the zeroth Landau level, n_G is the gate-induced carrier density, P is the percentage of the electron that contributes to the Landau levels, B is the magnetic field, ν is the filling number, and ϕ_0 is the unit magnetic flux. However, the fitting parameters can be unreliable and misleading with a two-parameter (P and n_0) fitting to only three levels ($C = 1$, $C = 2$, $C = 3$). An example is shown in **Figure. R3**. For this reason, we prefer not to present the fitting result in the Supplementary Materials.

Fig. R3. Landau fan fitting of Device 2.

3. In Fig. 2a, color code may be misleading because the red region looks $C = -1$ state or larger than $-1 h/e^2$ and

dark blue region around low magnetic field looks larger than $+1 h/e^2$. However, no experimental raw trace data is provided in main text and supplement. The readers cannot recognize what happens at low magnetic field and low nG region. It would great that the raw trace data of ρ_{yx} and ρ_{xx} for $nG = +0.5 \times 10^{12}$ and $-0.5 \times 10^{12} \text{ cm}^{-2}$ are shown in supplement. It is rather difficult to distinguish the values of ρ_{xx} and ρ_{yx} in the color map. Then, the authors explain what happens around there.

Response: Thanks for the good suggestion. **We added these linecuts of ρ_{xx} and ρ_{yx} in Supplementary Fig. 2 and revised the corresponding description in the main text.**

For general readers, the reason why insulator without surface state at antiferromagnetic condition at zero magnetic field should be also explained.

Response: We assume that the reviewer asked about why the QAH effect is absent in the AFM regime of our devices. Prior high-resolution ARPES studies have demonstrated the existence of the Dirac surface states in MnBi_2Te_4 crystal, confirming that the bulk MnBi_2Te_4 should be a topological insulator at zero magnetic field. However, for MnBi_2Te_4 thin films, the situation at the AFM state becomes complicated. As noted above, only a single device in Ref.15 reported the QAH effect under zero magnetic field. This result has not been reproduced yet. On the other hand, $C = 1$ Chern insulator state in the field induced FM state has been reported by several groups. Therefore, further studies are needed to clarify the topological properties of MnBi_2Te_4 thin films under zero magnetic field.

4, As for the appearance of $C = 2$ at high magnetic field and large nG in Fig. 2, probably two prerequisites should be fulfilled. One is location of Fermi energy. For example, at n_2 in Fig. 2c, did the exchange gap position shift against Fermi energy by applying magnetic field? Fermi energy should locate far from both the exchange gap at top and bottom surface states to appear $C = 2$, because if Fermi energy locates in the exchange gap at either surface, Chern number can only be 1 with LL formation at other surface. It seems reasonable that the shift of Fermi energy by applying electric field or dual gating. However, around n_2 in Fig. 2, the transition occurs from $C = 1$ to $C = 2$ by increasing magnetic field, which is one of main achievement in this study. It would be great that the authors clearly explain the mechanism for this transition.

Response: To understand the $C = 1$ to $C = 2$ transition in Fig. 2, we note that there are indeed ongoing theoretical efforts such as PRL127, 236402 (2021) (published on December 3, 2021, posted in arXiv (arXiv:2107.04752) after the submission of our paper), which we now added as a reference in the revised manuscript. This PRL paper proposes that for MnBi_2Te_4 under high magnetic field, the lowest Landau level, formed by Anderson localized state, together with QAH edge state, leads to a $C = 2$ state. Similar effects have also been studied in other material systems. For example, PRB 102, 165305 (2020) observed a $C = 2$ to $C = 1$ and then back to $C = 2$ transition, attributed to two species of holes in the system. In PRB 85, 195304 (2012), in a related magnetically doped quantum well (Mn, Hg)Te, a reentrant quantum Hall effect (transition from $C = 2$ to $C = 1$ due to linear Hall effect, and then enters $C = 2$ again as magnetic field increases) is predicted due to Mn ions' effective nonlinear Zeeman effect. Here, the observed transition from C

= 1 to $C = 2$ as the magnetic field increases in MnBi₂Te₄ could share the same origin as this quantum well system.

In the revised manuscript, we added the above discussion and cited PRB 85, 195304(2020) and PRL127, 236402 (2021).

In addition, the claim for the observation of $C = 3$ seems too strong because plateau in ρ_{yx} is narrow and difficult to recognize the accuracy of quantized value. The magnified result of ρ_{yx} and ρ_{xx} shown in supplement is helpful to understand it.

Response: Fig. R4 shows the evolution of $C = 3$ by taking the linecuts. We can see the $C = 3$ state developing there.

We added the derivative data and linecuts in Supplementary Fig. 2.

Figure R4. More Evidence of $C = 3$ state. **a**, Replot of Fig 2a for comparison. **b**, Derivative of (a) showing the emergence of $C = 3$ state. **c**, **d**, linecut of ρ_{xx} and ρ_{yx} at $\mu_0 H = 9.5T$ (c) and high doping level $n_G = 2.9 \times 10^{12} \text{cm}^{-2}$ (d). It shows a nearly quantized plateau where $\rho_{yx} \sim 0.31 h/e^2$ that corresponding to a dip of ρ_{xx} curve.

The asymmetry between electron and hole region may not be explained clearly in the present manuscript.

Response: Indeed, electron and hole asymmetry is appreciable in all reported experiments, including ours. The reasons are two-fold: i) MBT's band carries an electron-like topological number ($\rho_{yx} = -\text{sgn}(B) h/e^2$). When the first Landau level with $\nu = -1$ of the hole band develops, the behavior of the hole band will be like $C = 1 + (-1) = 0$, and an insulating state possibly develops. ii) MBT itself has electron-hole asymmetry, which is well examined in a high field experiment, e.g., Ref. 18.

We added the above discussion in the caption of Supplementary Fig. 4.

At zero magnetic field, the basic transport property should be shown in supplement such as ρ_{xx} as a function of n_G for all devices, because color map does not work to compare the values. The comparison with line data will evidence the reproducibility in this study. If the value of ρ_{xx} is rather high, of course the ρ_{xx} as a function of n_G at high field is alternatives.

Response: We thank the referee for pointing out a way to improve our data presentation. We now show the basic transport properties ρ_{xx} as a function of n_G in Supplementary Fig. 5. In Supplementary Figs. 6, we also provide both ρ_{xx} and ρ_{yx} vs. $\mu_0 H$ at various n_G . As the reviewer

suspected, ρ_{xx} is rather high ($\sim G$ ohm) at certain dopings, which makes the measurement not reliable for those doping levels.

We added requested data as Supplementary Fig. 5.

5, H_{c2} looks shifted in n_2 in Fig. 2f from in n_1 in Fig. 2e while H_{c1} is kept constant. The shift of H_{c2} can be detected by MCD? This shift may be related on the appearance of $C = 2$ at high magnetic field.

Response: In Fig. R5 below, we show RMCD data as a function of carrier density in a relatively high field. The shift of H_{c2} is not appreciable. We note that H_{c2} shows as a kink feature in transport, buried in the quantum oscillation feature in Fig. 2f and Fig.2e. So, extracting H_{c2} from a single trace of transport data is not reliable.

Figure R5. RMCD as a function of magnetic field at various carrier density (n) and displacement field (D).
Insert: Zoomed-in at high fields show that H_{c2} is independent of n and D .

While the H_{c1} variation with applying electric field is discussed in Fig. 4, the presented result is limited magnetic field region. The description of [The RMCD measurements exclude electric field tuning of the magnetic state as the origin of the electric control of the Chern insulator state.] may be not clear. Because the transition from $C = 1$ to $C = 0$ or 2 is discussed in this manuscript, the consistent discussion is necessary through overall in this manuscript.

Response: As discussed in the paper, the electric field tuning of the $C = 1$ state is only achieved in the cAFM state near H_{c1} , as revealed in Fig.3. We can achieve this electrical control because the exchange gap is small near the AFM to cAFM phase transition.

We clarified this point in Discussion section of the revised manuscript.

6, is the description of [Increasing $\mu_0 H$ increases the degeneracy of LLs] in page 3 accurate? Could the authors recognize that the $C = 1$ comes from LL formation or QAH exchange gap? This point seems not clearly discussed in the present manuscript. It would be great that the authors define the region for QAHE-driven $C = 1$ or LL-driven $C = 1$ if it exists.

Response: "Increasing $\mu_0 H$ increases the degeneracy of LLs" is a textbook explanation of the quantum Hall effect in a 2D electron gas system. The number of allowed states, $n_L = \frac{B\nu}{\phi_0}$, is proportional to magnetic field B . In our work, the Chern insulator nature of the $C = 1$ state in the low magnetic is supported by: 1) The shape of $C = 1$ area in the phase diagram. In a nonmagnetic and quadratic 2D electron gas with linear Landau level spectra, the shape of $C = 1$ area should follow a fan diagram that is defined by the $n_L = \frac{B\nu}{\phi_0}$, which is clearly not the case for our observation. 2) The formation of $C = 1$ has a clear coupling to magnetic states. 3) The transition from $C = 1$ to $C = 2$ as the magnetic field increases is inconsistent with the LL picture, showing that both states have distinct topological origins. 4) The $C = 1$ Chern insulator nature in the field induced FM state is well established in the literature (see Ref. 15-18). Although the mechanism that induces $C = 1$ state can be different, the QAHE-driven $C = 1$ or LL-driven $C = 1$ are adiabatically connected and has no physical phase boundary, as discussed in Ref.22 (Phys. Rev. B 101, 195433 (2020)).

7, Can the authors comment on the controllable energy range based on Fig. 4c? In the devices in this study, large value of charge carrier density is controlled roughly 1×10^{13} in Fig. 4. By applying dual gating, how large energy range can be controlled in Fig. 4c? Controllable range is much larger than the gap size?

Response: We thank the reviewer for raising this question. Converting the gate-induced doping to a real energy scale requires self-consistent calculation. However, this calculation needs knowledge of the density of states, dielectric constant, and disorder-related effects, which are not yet known at this stage for MnBi_2Te_4 .

Could the authors control only Fermi energy with keeping $D = D_{\text{opt}}$ condition? Naively, to observe $C = 2$, Fermi energy locates far from both gap at top and bottom.

Response: Yes, the dual gated device enables to tune the carrier density at fixed displacement field D . Figure 2 shows ρ_{yx} as a function of n and magnetic field at fixed D . We find that to observe $C = 2$, Fermi level is located outside the $C = 1$ gap.

It would be great that the authors address the controllable range of energy in the device. Then, it is further great that the comparison between the controllable range and gap size rough quantitatively.

Response: As mentioned above, we could not achieve quantitative information on the controllable energy range. To gain such information accurately, spectral-resolved techniques are required (e.g., STM/S, ARPES), which is beyond the scope of this work.

8, In Discussion part, the description of [the magnetic exchange gap increases. The electric field modulation of the Chern insulator state becomes weaker but is still feasible.] is not well organized. The origin for the increase of the magnetic exchange gap seem comes from the enhancement of exchange coupling owing to increase of out-of-plane magnetization. If so, the gap size is saturated

at H_c2 at ferromagnetic condition. This point should be described. The reason why the electric field modulation is weakened is not clearly discussed in anywhere in the present manuscript. It would be great that these descriptions are revised with explanation based on the experimental results.

Response: We modified the relevant discussion in the revised manuscript.

Technical comment

1, It is rather difficult to recognize the yellow line and region indicating $C = 3$ in Fig. 2a. Please change color.

Response: Done.

2, The color code of Fig. 3c seems confusing because blue region is widely shown. However, most of the blue region in r_{xx} corresponds to zero in r_{yx} in Fig. 3b. If so, it would be great that the addition of periodic contour line roughly every h/e^2 may be helpful to recognize. Please improve the color.

Response: The large blue region is a low resistive state with small but nonvanishing ρ_{xx} , when the sample is doped deeply into the conduction or valence bands. We revise the color code for clarification. We thank the referee for bringing up the idea of using a periodic contour line. However, the current contour has its physical meaning by marking the quantized area with $\rho_{yx} < 0.9 h/e^2$ and $\rho_{xx} < 0.1 h/e^2$. The remarkable single contour would help understand how the electric field controls the topological properties. We improved the color contrast to make the zero-resistance state more prominent.

3, In method section, device fabrication, please add a reference at [as previously reported].

Response: Done.

Response to Referee 3

In this manuscript, Cai et al systematically studied the electrical transport behaviors of the recently discovered intrinsic AFM TI $MnBi_2Te_4$. They found the Chern insulator state with quantized R_{yx} and zero R_{xx} appears at a low magnetic field of 3.5 T corresponding to a canted AFM state identified by RMCD measurements. By applying a magnetic field, the Chern insulator with $C = 1$ is driven into orbital quantum Hall states but with higher Chern numbers. In addition, the authors fabricated a dual-gated device to demonstrate the effects of electric field and carrier doping on manipulating the topological and magnetic properties of $MnBi_2Te_4$. This work stands in the continuation of efforts on the studies of the first van der Waals topological magnet $MnBi_2Te_4$ that has been proposed and demonstrated to host a rich variety of topological quantum phases, such as the Chern insulator and the axion insulator. Although there have already been relevant reports on the field-induced Chern insulator state in $MnBi_2Te_4$, the Hall quantization in most previous works is attributed to the formation of FM rather than canted AFM as the authors found in the experiment. The $C = 1$ to $C = 2$ phase transition indicates both the Chern insulator (quantum anomalous Hall

effect) and quantum Hall effect can coexist in MnBi₂Te₄, which is absent in magnetically doped TIs.

We thank the reviewer for his/her concise summary of our work and thoughtful comments below.

However, I find the claim of the canted AFM Chern insulator state unconvincing in its line of reasoning. More reproducible results are required to pin down whether this claim arises from accidental observation. Of particular note is that there are inconsistencies between the current data and the results in early works, which are never discussed in the manuscript.

Response: We would like to emphasize that the observation of canted-AFM Chern insulator state is reproducible. The original manuscript showed three devices with cAFM Chern insulator state with $C = 1$. We reproduce the $C = 1$ canted-AFM Chern insulator in an additional device, which are added to the revised Supplementary Materials (in total 4 devices). We hope that the observation in multiple devices removes the concern of reproducibility.

We thank the reviewer for their suggestion on comparing our data with previous reports. Ref 15-18 and 25 all reported a Chern insulator state in field-induced ferromagnetic states. Starting from this state, decreasing magnetic field introduces an FM to cAFM magnetic phase transition. This phase transition is of second-order type, which results in a continuous reduction of the exchange gap. During this process, one would expect that the Chern number remains one, as long as the gap larger than the thermal excitation energy. Actually, in some of the samples in Ref.17, a signature of the $C = 1$ state in the cAFM state near H_{c2} appears (Fig.3 in Ref 17). However, due to the lack of unambiguous measurements of the magnetic states (*e.g.* RMCD) and the limited experimental temperature, none of the previous reports identified this cAFM Chern insulator state. In particular, the appearance of $C = 1$ state associated with the AFM to cAFM phase transition is not identified. In addition, the electric field control of the cAFM state near the cAFM-FM phase boundary has not been realized.

In general, it is challenging to provide an accurate argument why others' experiments did not observe the canted-AFM Chern insulate state since details of others' experimental conditions are not known (basically, it is hard to interpret somebody else' experiment results). In addition, all experimental groups are using different crystal sources, while sample qualities are critical. Nevertheless, we speculate possible reasons why previous studies did not observe cAFM Chern insulator state:

- i) Low sample quality. The disorder can close the gap in cAFM state. We observed a device quality improvement when we made all our solvent anhydrous. We also see generally higher carrier mobility in dual gated devices made of stencil mask, in which the devices have never been exposed to any solvent. After submitting this manuscript, we note that the stencil mask method is adopted in a paper appearing in *Nature* **595**, 521 (2021), which produces a device with similar mobility $\sim 1.1 \times 10^3 \text{ cm}^2\text{V}^{-1}\text{s}^{-1}$.
- ii) Unintentional doping of the top surface. Without a top gate, the top surface of the device is exposed to air or PMMA, which are low-quality dielectrics. This could lead to high doping to the top surface and thus miss-align the already small exchange gaps

in the cAFM state between the bottom and top layers. Supplementary Fig. 3 shows how tuning the top gate changes the quantization field, supporting our understanding.

- iii) Higher measurement temperature. In this study, we mainly present our transport study in 50-300mK. However, the previous research mainly focuses on transport in ~ 2 K and above. Considering the small exchange gap in the cAFM state, in particular near H_{c1} , where the exchange gap is small, the millikelvin temperature is needed to overcome the thermal excitation.

The discussion on the physical origins is also too simple. Importantly, I find the surface exchange gap picture can even give rise to contradictory expectations to the experiments. For the $C = 1$ Chern insulator to $C = 2$ transition, the authors claimed the former one is due to the exchange gap and the latter is due to Landau levels. However, a Chern insulator state should have its Fermi level exactly located in the gap. In the quantum Hall state, the Fermi level should be outside the bandgap (in the conduction band for the present situation) so that the Landau levels play a role. Generally, these two topological states exist exclusively and cannot be switched between each other simply by applying a magnetic field. The theoretical discussion should be modified.

Response: To understand the $C = 1$ to $C = 2$ transition in Fig. 2, we note that there are indeed ongoing theoretical efforts such as PRL127, 236402 (2021) (published on December 3, 2021, posted in arXiv (arXiv:2107.04752) after the submission of our paper), which we now added as a reference in the revised manuscript. This PRL paper proposes that for MnBi_2Te_4 under high magnetic field, the lowest Landau level, formed by Anderson localized state, together with QAH edge state, leads to a $C = 2$ state. Similar effects have also been studied in other material systems. For example, PRB 102, 165305 (2020) observed a $C = 2$ to $C = 1$ and then back to $C = 2$ transition, attributed to two species of holes in the system. In PRB 85, 195304 (2012), in a related magnetically doped quantum well (Mn, Hg)Te, a reentrant quantum Hall effect (transition from $C = 2$ to $C = 1$ due to linear Hall effect, and then enters $C = 2$ again as magnetic field increases) is predicted due to Mn ions' effective nonlinear Zeeman effect. Here, the observed transition from $C = 1$ to $C = 2$ as the magnetic field increases in MnBi_2Te_4 could share the same origin as this quantum well system.

In the revised manuscript, we added the above discussion and cited PRB 85, 195304(2020) and PRL127, 236402 (2021).

In addition to the magnetic field tuned topological phase transition, the role of the electric field in adjusting the relative energies of the exchange gaps is also questionable. I make the statement based on the phase diagram in supplementary Fig. 2. Tuning the relative energies of the exchange gap will shrink the window of V_g for observing the $C = 1$ phase. However, judging from the experimental data, not only the shape but also the area of the $C = 1$ phase is insensitive to V_g .

Response: We would like to clarify that we have demonstrated the electric field tuning of the cAFM Chern insulator state near the cAFM-FM phase boundary, but not the Chern insulator state away from the boundary as in the field induced FM state. In cAFM state near the cAFM-FM phase boundary, the exchange gap size is expected to be small, which is feasible for the electric field

tuning, as shown in Fig. 3. Away from cAFM-FM phase boundary, the E field tuning of the C=1 state is negligible, as the reviewer pointed out.

To put it in the least way, regardless of whether the picture can account for the data, whether there are surface states in FM MnBi₂Te₄ remains a question. There is no doubt AFM MnBi₂Te₄ is a TI. But for its FM state, many theories have pointed out switching magnetic structure from AFM to FM can lead to a Weyl semimetal (PRL 107, 127205 (2011), PRL 122, 206401 (2019) and Sci. Adv. 5, eaaw5685 (2019)). In this situation, the concept of surface state, as well as the exchange gap, no longer exists. Several experimental works also provide evidence for the field-induced Weyl semimetal state, as shown in the two early works (arXiv:2001.08401 and arXiv:2002.10683). Interestingly, the Weyl semimetal picture gives a more comprehensive explanation of the presence of quantum Hall state in MnBi₂Te₄ Chern insulator.

Response: We thank the referee for the enlightening discussion about the Weyl semimetal picture. We are aware of the recent theory work on field-induced Weyl semimetal state in 3D bulk crystal of MnBi₂Te₄. If the Weyl semimetal picture is correct, our thin MnBi₂Te₄ film can be viewed as the Weyl semimetal states in the 2D quantum-confined limit (i.e., quantum well) with a gapped phase. This quantum well picture is adopted in recent Nat. Communications 12, 4647 (2021) (arXiv:2001.08401 mentioned by the referee) in the field-induced FM state. However, in the cAFM phase, the resulting quantum well is still a cAFM Chern insulator with $C = 1$ that can be viewed with the top and bottom surface states. In our thin-film experiment (5 to 7 SLs) with <10 nm scale, the modeled effective z-direction momentum \widetilde{k}_z can be hardly aligned to the Weyl point, in which case the system can be understood as an effective ‘2D Weyl semimetal’. We agree with the referee that in FM state, there could be more physics going on. We also think the Weyl semimetal picture is interesting but needs more direct experimental evidence to support such a picture and requires further systematic studies.

We have briefly discussed the Weyl semimetal picture in the field induced FM state in the revised manuscript.

In spite of all these concerns, I still find the topic in the manuscript is actual, and the reported experimental phenomena are interesting enough. I believe it would draw extensive attention from researchers in the fields of topological state of matter and material science. However, at the current stage, I cannot recommend this article without significant revisions. I suggest the authors provide more supportive data and give more reasonable explanations to their observations.

Please consider the following questions and comments:

1. I find there are some inconsistencies with the RMCD data for the three 7-SL devices. The authors should explain why samples with the same thickness have different H_{c1} .

Response: The reviewer is right that there is a slight difference (~ 0.2 T) of H_{c1} (about 3.8T) between the three 7-SL devices. As shown in Fig. 4, H_{c1} is doping-dependent. In single gated devices, the doping condition of the top surface varies from sample to sample, leading to variation of H_{c1} in the RMCD data.

2. The authors compared the RMCD signals with the transport data to identify the $C = 1$ Chern insulator phase is in canted AFM state (Fig. 1). Unfortunately, this comparison is inaccurate. Because both the transport and magnetic properties of MnBi_2Te_4 are temperature dependent. A reasonable comparison should be made at the same temperature, such as at 2 K.

Response: As we have already pointed out in the original manuscript, there is a slight difference of H_{c1} between the transport (3.6T saturation field of ρ_{yx}) and RMCD data (3.8T for the RMCD to have a quick rise), which is mainly from the different temperature. However, this small difference does not change the conclusion - the observation of $C = 1$ Chern insulator state in the cAFM state since the magnetic field range of the cAFM state is from 3.8T to 7.2T. The small difference (0.2T) of H_{c1} is negligible compared to this large overlap of the field range between the quantized ρ_{yx} and cAFM state. On the other hand, since the exchange gap is small, particularly near the cAFM-FM phase boundary, the Chern insulator state will not appear near the cAFM-FM phase boundary at 2K. So the comparison of the transport and RMCD data at 2K maybe not meaningful.

3. "Chern insulator phase in the canted AFM state" is an important claim. However, I find this conclusion is drawn mainly from the results of Device 1 (Fig. 1). Considering the large discrepancies of H_{c1} in the samples and the lack of comparisons between RMCD and transport data in Devices 2 and 4, I cannot rule out the possibility that this conclusion arises from cherry-picked data. I make the statement also based on comparing Fig. 1 with previously published results (Science 367, 895 (2020), Nat. Mater. 19, 522 (2020), Nat. Sci. Rev. 7, 1280 (2020), Nano Lett. 21, 2544 (2021)). In these reports, most groups found the Chern insulator state appears in the FM state rather than in the canted AFM state. Even in the Nano Letter work by some of the authors, the samples do not show quantitation until entering the FM state. Although I believe this finding is interesting, more reproducible data are required to convince general readers. What about the transport data for Device 4? How many samples are measured and does the Chern insulator state always appears in the canted AFM state?

Response: We thank the reviewer for stressing the importance of reproducibility. We share the same philosophy. We have compared our results with previous reports in the response to the first comment. So, we refer the reviewer to the response there.

For devices 2 and 3, although RMCD data cannot be taken due to the metal top gate, we can clearly see the sharp phase boundary from the quantized ρ_{yx} and the near vanishing ρ_{yx} in the AFM state. Besides, the phase boundary for the quantized ρ_{yx} is located near the H_{c1} (the exact value can vary slightly between samples and temperature, see our RMCD data in Nano Lett. 21, 2544-2550 (2021), Ref.17), and at a much lower magnetic field (all saturation field of $\rho_{yx} < 4\text{T}$) than the critical field needed for the formation of the magnetic field induced FM state ($\sim 7\text{T}$). Using Device 2 in Fig. 2 as an example, the magnetic field range of the $C = 1$ state before the formation of FM state is about 3.6T to 7T, where the magnetic state must be in the cAFM. In addition, the shape of the $C = 1$ region in the phase diagram is very similar in the dual gate Devices 2 and 3 compared to the single gated Device 1 (See Fig. R1 which is also added in the Supplemental Materials). Therefore, there is no doubt that the Chern insulator state is formed in the cAFM state in Devices 2 and 3. During

the preparation of the response, we have measured an additional single gated device for simultaneous transport and RMCD measurements. As expected, the data reproduces the $C = 1$ cAFM state.

We added the data of the additional single gated device in Supplementary Fig. 7.

4. "Unlike the $C = 1$ phase, the contours of $C = 2$ and 3 phases are linearly dependent on magnetic field $\mu_0 H$ (Fig. 2a). This implies that the $C = 2$ and 3 states are a result of the Landau level (LL) formation coexisting with edge state from band topology (ref. 21)". The linearity of the $C = 2$ and $C = 3$ phases is only marked by the manually added dashed lines. I suggest the authors show the first derivative data. In fact, the $C = 3$ phase in Fig. 2a is not clear. Even in Figs. 2b and 2c, it is difficult to identify the $\rho_{yx} = h/3e^2$ and $\rho_{yx} = 0$ quantum Hall plateau. It seems the claim of $C = 3$ quantum Hall state is an overstatement. In addition, could the authors estimate the sample mobility so that readers may judge the relevance of the Landau levels to the high Chern number state?

Response: We thank the referee for this excellent suggestion on data presentation. Here we show the first derivative of Fig. 2a in Fig. R4 (now included in Supplementary Fig. 2c), which identifies the phase space for $C = 1, 2,$ and 3 states. Fig. R4 also shows a linecut to further support the $C = 3$ state.

Figure R4. More evidence of $C = 3$ state. **a**, replot as Fig 2a for comparison. **b**, derivative of **(a)** showing the emergence of $C = 3$ state clearly. **c**, linecut of ρ_{xx} and ρ_{yx} at very high doping level $n_G = 2.9 \times 10^{12} \text{cm}^{-2}$. It shows a nearly quantized plateau where $\rho_{yx} \sim 0.31h/e^2$ that corresponding to a dip of ρ_{xx} curve.

We added the cuts to support the $C = 3$ state in Supplementary Fig. 2 and presented the sample mobility in Supplementary Table 1.

Device type	Device # in the text	Thickness (# of SLs)	cAFM Chern insulator?	Mobility ($10^3 \text{cm}^2 \text{V}^{-1} \text{s}^{-1}$) (50mK, $\sim 3 \times 10^{12} \text{cm}^{-2}$)
Single gated device	1	7	Y	0.2
	5	8	Y	0.9
Dual gated device	2	7	Y	1.7-3
	3	6	Y	1.5-2

Table R1. A list of the transport devices. To obtain electron mobility, we gate the device to be far away from anomalous Hall region and extract Hall carrier density and mobility by fitting the linear Hall effect with respect to small magnetic field.

5. As I have mentioned at the beginning, one major drawback of the theoretical picture (surface exchange gap + Landau levels) is that it only provides a possible explanation to $C = 2$ quantum Hall state but cannot explain why it could arise from a $C = 1$ Chern insulator. For fixed Fermi level position, depending on in the exchange gap or in the conduction band, the system should be either a Chern insulator or a quantum Hall state (if Landau levels form).

Response: This is the same comment as the one started with “The discussion on the physical origins is also too simple.” Please see our response there.

6. Regarding the opinion "the Landau level (LL) formation coexisting with edge state from band topology", it has already been demonstrated experimentally by transport studies of MnBi_2Te_4 flakes (arXiv:2001.08401). Differently, in this arXiv work, the Chern insulator to quantum Hall state transition is explained on the basis of Weyl semimetal picture rather than magnetic TI. In my eyes, depending on the number of Landau levels in the conduction subband, the Weyl model can naturally explain the $C = 1$ Chern insulator to $C = 2$ quantum Hall phase transition observed in the present manuscript.

Response: We thank the referee for pointing out a possible understanding of our data. In Nat. Communications 12, 4647 (2021) (arXiv:2001.08401 mentioned by the referee, now as Ref.18 in the maintext), they observed the Chern insulator to quantum Hall state transition only in FM state and at a much higher magnetic field (20T compared to as low as 8T in our data). In addition, in the FM state, we clearly establish doping dependence. We feel like before going to the Weyl metal picture, whose validity still needs further experimental evidence, a simple quantum Hall with a mixture of edge states can already explain our data. In addition, as we have mentioned, the PRL127, 236402 (2021) (published on December 3, 2021, posted in arXiv (arXiv:2107.04752) after the submission of our paper) proposes that for MnBi_2Te_4 under magnetic field, the lowest Landau level, formed by Anderson localized state, together with QAH edge state can lead to a $C = 2$ state. On the other hand, the Weyl metal picture cannot explain the observed $C = 1$ in cAFM to $C = 2$ state transition in the FM state in a low magnetic field.

We have revised the text accordingly.

7. "The behavior of ρ_{yx} and ρ_{xx} suggests that the $C = 1$ state can be switched on and off by an external electric field". This statement is inaccurate. The experiment was carried near the phase boundary. A lot of factors can easily smear out the Chern insulator quantization. The deviation of ρ_{yx} from h/e^2 to an unquantized value and the appearance of ρ_{xx} do not mean the Chern insulator is turned off but only indicate the presence of diffusive conduction channels. A real "switch on and off" refers to topological phase transition with gap close and reopen. Therefore, the "electric field control of the band topology" is not realized.

Response: This is an excellent point. We agree with the referee that switching on and off the edge state should correspond to switching between two gapped states with an MTI or a normal insulator.

We change the statement "edge state can be switched on and off" in the manuscript to "dissipationless transport can be switched on and off". We think this is a more accurate description and thank the referee for the suggestion.

REVIEWER COMMENTS

Reviewer #2 (Remarks to the Author):

Dear Authors,

I appreciate the authors for appropriate revisions and responses.

Though detail responses are provided, the feasibility of the appearance of $C = 2$ is still unclear. Therefore, I would suggest to add the data of temperature dependence for consideration by general readers. In addition, I also suggest to add the explanation for the real space picture of QAHE and QHE for $C = 1$ and 2. These data and descriptions are quite beneficial for the consideration and judgement by the general readers.

1, I strongly recommend adding a data set for temperature dependence of raw_xx and raw_yx at $B = 0$ T and $C = 1, 2, 3$ at different magnetic field and electric-field applied. Across Curie temperature at ferromagnetic condition and Neel temperature at AFM condition, temperature dependence of raw_xx and raw_ryx would be distinguished each quantum phases because the exchange gap and Landau splitting gap would clearly present different temperature dependence.

These data set is quite beneficial for considering the feasibility of the appearance of $C = 2$ by QAHE and LL QHE. For general readers, such sufficient results for consideration should be provided because referee 3 is also skeptical for the observation and discussion of origin for $C = 2$ and 3.

2, QAHE expectedly appears in 2D systems. In the present main text and discussion, it is not explicitly defined this point. Did QAHE occur at only surface state? Did QHE by Landau level formation occur at bulk region?

Referee 3 also pointed out the difficult understanding of the Fermi energy position. Therefore, I strongly recommend adding the description about the surface state and bulk region. When surface state reaches QAHE, the surface state does not contribute to Landau level formation because Fermi energy is located at the exchange gap of the surface state. By applying magnetic field, the Landau level formation may occur at bulk region. Moreover, it is also important whether the bulk conduction is 2D or 3D for QHE. Please make this point clear by adding the description in the main text before publication.

Reviewer #3 (Remarks to the Author):

First, I would like to clearly state that the current manuscript meets the standards for publication in Nature Communications.

Now, having carefully read the paper, I take the liberty of adding my own comments. There is no doubt that to have a $C = 2$ Chern insulator phase, Landau levels are indispensable. However, just as I have mentioned in the first report, it remains unclear whether the Landau levels arise from the TI surface states or the quantum well states (Weyl semimetal picture). In the revised manuscript, the authors added a sentence "as the magnetic field increases, the magnetic exchange gap becomes larger as well due to the increase of out-of-plane magnetization". It is expectable within the framework of TI picture. However, the current work does not provide any experimental evidence that the exchange gap indeed becomes larger in magnetic field. Most importantly, the gap fitting in the Science work clearly shows an opposite result, the gap size is reduced in magnetic field. It deserves future studies to determine the exact topological ground state in FM $MnBi_2Te_4$. But at the current stage, the authors should revise the inaccurate description.

We sincerely thank all referees for their further comments on our manuscript. In the following, we provide a point-to-point response to their comments. Major changes are shown in red in the revised manuscript.

Response to Referee 2

I appreciate the authors for appropriate revisions and responses. Though detail responses are provided, the feasibility of the appearance of $C = 2$ is still unclear. Therefore, I would suggest to add the data of temperature dependence for consideration by general readers. In addition, I also suggest to add the explanation for the real space picture of QAHE and QHE for $C = 1$ and 2. These data and descriptions are quite beneficial for the consideration and judgement by the general readers.

1. I strongly recommend adding a data set for temperature dependence of raw_{xx} and raw_{yx} at $B = 0$ T and $C = 1, 2, 3$ at different magnetic field and electric-field applied. Across Curie temperature at ferromagnetic condition and Neel temperature at AFM condition, temperature dependence of raw_{xx} and raw_{yx} would be distinguished each quantum phases because the exchange gap and Landau splitting gap would clearly present different temperature dependence.

These data set is quite beneficial for considering the feasibility of the appearance of $C = 2$ by QAHE and LL QHE. For general readers, such sufficient results for consideration should be provided because referee 3 is also skeptical for the observation and discussion of origin for $C = 2$ and 3.

We thank the referee for the proposal on taking temperature dependence data to distinguish the nature of $C = 1, 2, 3$ states. We argue here that temperature-dependent measurement is not desirable for distinguishing these quantum states in MnBi_2Te_4 .

1. A discrete set of $R(T)$ curves on different states is not enough to distinguish the origins of higher Chern number states – the different gapped states will appear as a transition on the measured thermally activated gap function $\Delta(T)$. The $R(T, B)$ map is also not enough since the AFM-FM-Paramagnetic signal would mix up with the signal from different phases. For instance, in Ref. 15 (Science 367, 895-900 (2020)), a small range of temperature-dependent data shows that extracted $\Delta(T)$ shows a peak near canted AFM to FM transition.
2. Gated MnBi_2Te_4 also shows temperature-dependent charge neutrality point, as clearly witnessed by temperature-dependent data in Ref. 17 (Nano Lett. 21, 2544-2550 (2021), Figure 4).
3. Theoretically, for the $C = 2$ state, a gapped state with $\sigma_{xx} = \sigma_0 e^{\Delta/2k_B T}$, temperature-dependent data in fixed magnetic fields won't show difference whether it's purely from Landau level or coexistence of QAH and QH, *e.g.*, shown in Fig. 3's insert of Ref. 34 (PRL 144 187201 (2015)).

All these factors will make temperature-dependent data insufficient to understand the origin of the high Chern number state. Below, we list the collective experimental facts that support the assignment of $C = 1$ as Chern insulating state, and $C = 2, 3$ as quantum Hall states:

1. The $C = 1$ state in the cAFM phase has a nonlinear shape in the $n - \mu_0 H$ phase diagram, which shows that the $C = 1$ state is a Chern insulating state arising from band topology in the cAFM phase. On the contrary, the $C = 2$ and $C = 3$ states as a function of doping n show linear dependence in $\mu_0 H$, as shown in Fig. R1.
2. The $C = 0$ to $C = 1$ topological phase transition accompanies the AFM to cAFM magnetic phase transition. This is another evidence that the $C = 1$ in the cAFM phase is related to an opening of the exchange gap.
3. The transition from $C = 1$ state to $C = 2$ state as the magnetic field increases demonstrate a distinct topological origin between the two. The mechanism could be attributed to the nonlinear Zeeman effect, as studied in Ref. 23, or the Anderson localization mechanism studied in Ref. 24 (discussed in detail in the previous response).
4. The $C = 1$ state can be observed in devices with low mobility $\mu \sim 200 \text{ cm}^2 \text{V}^{-1} \text{s}^{-1}$ (D1 in this study, and Ref.16), which cannot be explained by Landau level formation. With the sample of increased mobility to around $3000 \text{ cm}^2 \text{V}^{-1} \text{s}^{-1}$, the $C = 2$ state and $C = 3$ state appear in the high magnetic field and electron doping.

Fig. R1. Phase diagram reveal by ρ_{yx} as a function of $n - \mu_0 H$. Dashed lines are guide-to-the-eye for the linear shape of $C = 2$ (Landau level index $n_{LL}=1$) and $C = 3$ ($n_{LL}=2$).

2. QAHE expectedly appears in 2D systems. In the present main text and discussion, it is not explicitly defined this point. Did QAHE occur at only surface state? Did QHE by Landau level formation occur at bulk region?

Referee 3 also pointed out the difficult understanding of the Fermi energy position.

Therefore, I strongly recommend adding the description about the surface state and bulk region. When surface state reaches QAHE, the surface state does not contribute to Landau level formation because Fermi energy is located at the exchange gap of the surface state.

We respectfully disagree with ‘When surface state reaches QAHE, the surface state does not contribute to Landau level formation because Fermi energy is located at the exchange gap of the surface state’.

In the standard understanding of QH, the Landau level is developed from the band edge and form a fan-like pattern. However, when the band carries a non-zero Chern number, this non-zero Chern number will be added into the Landau level index. Details are discussed in Ref. 22 & 24. In Ref. 22 (PRB 101, 195433 (2020)) - by only considering the surface state’s effective Hamiltonian, Figure 1c shows that if the band carries Chern number, Landau index is distributed as 1, 0, -1 (in gap), -2, which deviates from standard Landau fan 1, 0 (in gap), -1 for a topologically trivial sample.

Therefore, when the surface state reaches QAHE and Fermi energy is located in the first Landau level, the surface state manifests itself by adding Chern number one to the Landau level index.

When the Fermi energy is at the edge of the surface state gap, Ref. 23-24 explain that nonlinear Zeeman effect or Anderson localized state could lead to $C = 1$ (surface gap) to $C = 2$ (Landau level and Chern number contributed by surface state) transition, which is addressed during last revision.

By applying magnetic field, the Landau level formation may occur at bulk region. Moreover, it is also important whether the bulk conduction is 2D or 3D for QHE. Please make this point clear by adding the description in the main text before publication.

We are confused about ‘*whether the bulk conduction is 2D or 3D for QHE*’. We assume the question is about the origin of Landau level.

As in a magnetic topological insulator (MTI), Chern number can be considered as the equal contribution from the top and bottom surfaces, with Landau level index added to each half. That leads to the theoretical understanding in Figure 4 of Ref. 15 (Science 367, 895-900 (2020)). However, extra complexity appears if we hypothetically consider FM MnBi_2Te_4 as a confined Weyl semimetal, where the quantum well states originated from the bulk band could serve as the same topological band for the top (bottom) surface band of an MTI (Ref. 18 Nat. Commun. 12, 1-8 (2021)). In this sense, the quantum well states would not be spatially originating from any surface of the sample but bulk. Novel spatial distribution of the edge state may appear at high magnetic fields, as discussed in PRL 125, 036602 (2020), where the bulk conduction could be 3D. Our current data at relatively low fields could not distinguish the different possible theoretical pictures in FM states, as we have already mentioned in the manuscript, but it does not affect any claim of our observation of cAFM Chern insulator with its electric field control. However, we emphasize that using Weyl semimetal picture in thin flakes needs extra caution since it is not well established even in the bulk crystal MnBi_2Te_4 .

We have added the above discussion to the revised maintext.

Response to Referee 3

First, I would like to clearly state that the current manuscript meets the standards for publication in Nature Communications.

We thank the referee for the positive assessment of our work.

Now, having carefully read the paper, I take the liberty of adding my own comments. There is no doubt that to have a $C = 2$ Chern insulator phase, Landau levels are indispensable. However, just as I have mentioned in the first report, it remains unclear whether the Landau levels arise from the TI surface states or the quantum well states (Weyl semimetal picture). In the revised manuscript, the authors added a sentence “as the magnetic field increases, the magnetic exchange gap becomes larger as well due to the increase of out-of-plane magnetization”. It is expectable within the framework of TI picture. However, the current work does not provide any experimental evidence that the exchange gap indeed becomes larger in magnetic field. Most importantly, the gap fitting

in the Science work clearly shows an opposite result, the gap size is reduced in magnetic field. It deserves future studies to determine the exact topological ground state in FM MnBi₂Te₄. But at the current stage, the authors should revise the inaccurate description.

We thank referee for their insights. Our understanding is that ‘the magnetic exchange gap becomes larger’ happens in the cAFM phase from H_{C1} to H_{C2} , supported by the reduced electric field control as the magnetic field increases. This picture should be limited to cAFM phase. We agree with the referee that the origin of Landau level formation in FM state deserves further studies in the future. As mentioned in the previous response, the Weyl semimetal picture in bulk MnBi₂Te₄ has not been fully established experimentally. So, its usage in explaining thin flake data needs extra caution.

We have revised the corresponding discussion in the manuscript.